# Infection fatality rate of SARS-CoV2 in a super-spreading event in Germany

Hendrik Streeck [1,2✉], Bianca Schulte[1,2], Beate M. Kümmerer[1,2], Enrico Richter[1,2], Tobias Höller[3], Christine Fuhrmann[3], Eva Bartok [2,4], Ramona Dolscheid-Pommerich[2,4], Moritz Berger[5], Lukas Wessendorf[1,2], Monika Eschbach-Bludau[1,2], Angelika Kellings[3], Astrid Schwaiger[6], Martin Coenen[3], Per Hoffmann [7], Birgit Stoffel-Wagner[2,4], Markus M. Nöthen[7], Anna M. Eis-Hübinger[1,2], Martin Exner[8], Ricarda Maria Schmithausen[8], Matthias Schmid[5] & Gunther Hartmann [2,4✉]

A SARS-CoV2 super-spreading event occurred during carnival in a small town in Germany. Due to the rapidly imposed lockdown and its relatively closed community, this town was seen as an ideal model to investigate the infection fatality rate (IFR). Here, a 7-day ser-oepidemiological observational study was performed to collect information and biomaterials from a random, household-based study population. The number of infections was determined by IgG analyses and PCR testing. We found that of the 919 individuals with evaluable infection status, 15.5% (95% CI:[12.3; 19.0%]) were infected. This is a fivefold higher rate than the reported cases for this community (3.1%). 22.2% of all infected individuals were asymptomatic. The estimated IFR was 0.36% (95% CI:[0.29; 0.45%]) for the community and 0.35% [0.28; 0.45%] when age-standardized to the population of the community. Participation in carnival increased both infection rate (21.3% versus 9.5%, $p < 0.001$) and number of symptoms (estimated relative mean increase 1.6, $p = 0.007$). While the infection rate here is not representative for Germany, the IFR is useful to estimate the consequences of the pandemic in places with similar healthcare systems and population characteristics. Whether the super-spreading event not only increases the infection rate but also affects the IFR requires further investigation.

[1] Institute of Virology, University Hospital, University of Bonn, Bonn, Germany. [2] German Center for Infection Research (DZIF), Partner Site Bonn-Cologne, Braunschweig, Germany. [3] Clinical Study Core Unit, Study Center Bonn (SZB), University Hospital, University of Bonn, Bonn, Germany. [4] Institute of Clinical Chemistry and Clinical Pharmacology, University Hospital, University of Bonn, Bonn, Germany. [5] Institute for Medical Biometry, Informatics and Epidemiology, University Hospital, University of Bonn, Bonn, Germany. [6] Biobank Core Unit, University Hospital, University of Bonn, Bonn, Germany. [7] Institute of Human Genetics, University Hospital, University of Bonn, Bonn, Germany. [8] Institute for Hygiene and Public Health, University Hospital, University of Bonn, Bonn, Germany. ✉email: hendrik.streeck@ukbonn.de; gunther.hartmann@ukbonn.de

The SARS-CoV-2 coronavirus, the causative agent of the respiratory disease COVID-19, has affected almost every country worldwide[1]. One of the reasons for its rapid spread is its ability to transmit before becoming symptomatic, as has been reported for ~40% of SARS-CoV-2 transmission events[2,3]. As the COVID-19 pandemic continues to grow in extent, severity, and socio-economic consequences, its fatality rate remains unclear. Most estimates of the CFR (case fatality rate) are based on cases detected through surveillance and calculated using crude methods, giving rise to widely variable estimates of CFR by country as outlined by the WHO[1]. Since SARS-CoV-2 infection presents with a broad spectrum of clinical courses, from asymptomatic to fatal, cases with mild to moderate symptoms including sore throat, dry cough, and fever are often left undiagnosed[3–7]. In addition, different PCR-testing capacities and regulations have contributed to the variability of reported CFRs. As a consequence, epidemiological modeling is currently associated with a large degree of uncertainty. However, valid epidemiological modeling is urgently needed to design the most appropriate prevention and control strategies to counter the pandemic and to minimize collateral damage to societies.

Unlike the CFR, the infection fatality rate (IFR = number of deaths from disease/number of infected people) includes the whole spectrum of infected individuals, from asymptomatic to severe. The IFR is recommended as a more reliable parameter for evidence-based assessment of the SARS-CoV-2 pandemic[4] (Center for Evidence-Based Medicine, CEBM in Oxford). The IFR includes infections based on both PCR testing and virus-specific antibodies. PCR testing allows the inclusion of active infections before seroconversion into IFR-calculation. In addition, testing for virus-specific antibodies also includes past infections and those with mild and moderate disease courses, which do not tend to be captured and documented by PCR testing alone. Notably, ELISA tests for a reliable serological analysis of SARS-CoV-2-specific antibodies (specificity higher than 98%) became available only recently. However, the reliability of serological analysis is also strongly dependent on seroprevalence.

We chose to perform a seroepidemiological study in the German community of Gangelt, where, due to a super-spreading event, 3.1% of the population were officially reported to be SARS-CoV-2 PCR positive at the time of the study. In this community, carnival festivities around February 15, 2020 were followed by a massive outbreak of SARS-CoV-2 infections. Strict measures were immediately taken to slow down further spreading of the infection. Given its relatively closed community with little tourism and travel, this community was identified as an ideal model to better understand SARS-CoV-2 spreading, prevalence of symptoms, as well as the IFR. The study presented here was designed to determine the total number of infected and the IFR. In addition, the spectrum of symptoms, as well as associations with age, sex, household size, co-morbidities and participation in carnival festivities, were examined.

## Results

**Study setting and participants.** In the German community of Gangelt (12,597 inhabitants, January 1, 2020), a super-spreading event (carnival festivities around February 15, 2020), was followed by numerous measures starting February 28 (shutdown) to limit the further spread of infections (Fig. 1a). This local infection hotspot was closely monitored by health authorities, and a high PCR test rate revealed an increase in officially reported cases, with a maximum around March 13 when 85 individuals tested PCR positive for SARS-CoV-2 in a 4-day period. Numbers declined afterwards down to 48 PCR-positive cases officially reported during the 7-day period of the present study (March 31–April 6),

not counting the 33 new SARS-CoV-2 PCR positives detected by this study. The total number of officially reported PCR positives on April 6 was 388, also excluding the 33 PCR positives of this study. By the end of the 7-day period, a total of 7 SARS-CoV-2-positive individuals had died in the community of Gangelt since the super-spreading event (average age 80.8 years, sd ± 3.5 years). In January, February, and March 2020, a total of 48 people died in Gangelt. At the start date of data and material acquisition of the study, 340 PCR positives were reported in the community, which is 2.7% of the population.

For the study, 600 adult persons with different surnames in Gangelt were randomly selected and were asked to participate together with all household members. Nine-hundred eighty-seven individuals were seen in the local study acquisition center in a community school, and 20 individuals were visited in their homes due to age or limited mobility. Complete information from both pharyngeal swabs and blood samples was available for 919 study participants living in 405 households (Fig. 1b). The demographic characteristics of the study participants, including age, sex, and the number of participants living in the same household, are summarized in Table 1. The comparison of age groups in the study population to the community of Gangelt, to the state of North Rhine-Westphalia (NRW), and to Germany is illustrated in Supplementary Fig. 1. The age group 65 years and older is overrepresented in the study cohort as compared to Gangelt, NRW or Germany (Supplementary Fig. 1). Characteristics of the 88 study participants who were not evaluable for infection status, mainly children due to lack of biomaterials, are provided in the Supplementary Table 1. Details on power calculations, statistical analysis and handling of missing values are given in the Methods section.

**Number of SARS-CoV-2-infected and infection fatality rate (IFR).** The analysis of IgA and IgG levels measured in plasma samples of all study participants by ELISA (Euroimmun) showed a positive correlation ($r = 0.778$, CI 95%: [0.751–0.802]: Fig. 2a). While 18.50% of all study participants were found to be IgA positive, 13.60% were IgG positive (Fig. 2b). Statistical correction for sensitivity and specificity of the ELISA resulted in a corrected value of 10.63% [7.48%; 13.88%] for IgA and 14.11% [11.15%; 17.27%] for IgG (Fig. 2b). Based on these data, a "seropositive" study participant was defined as being positive for IgG (Fig. 2b, c). The neutralization activity of IgG-positive plasma samples was analyzed using a microneutralization assay combined with a plaque reduction neutralization test. Results are shown in Fig. 2d, e. Of the 919 participants of the study, 33 were tested positive (PCR$_{new}$: 3.59%). Twenty-two study participants reported that they had had a SARS-CoV-2-positive PCR test in the past (PCR$_{rep+}$: 2.39%). The combination of serology (non-corrected IgG values) and past and present PCR testing yielded a total number of 138 study participants (15.02%) that had been previously or were at that time point infected by SARS-CoV-2 as illustrated in Fig. 2c. The inclusion of IgG values corrected for sensitivity and specificity in the calculation resulted in an estimated 15.53% [12.31%; 18.96%] cumulative SARS-CoV-2-infected.

To determine the infection fatality rate (IFR), the estimated infection rate of 15.53% in the study population was applied to the total population in the community ($n = 12,597$) yielding an estimated number of 1956 [1,551; 2,389] infected people. With seven SARS-CoV-2-associated deaths, as reported to the authors by the local administration, the estimated IFR was $7/1956 = 0.36\%$ (95% CI: [0.29%; 0.45%] for Gangelt, [0.17%; 0.77%] when accounting for uncertainty in the number of recorded deaths) (Fig. 3a) at the end of the acquisition period. The age-standardized IFR for Gangelt (using the proportions in Supplementary Fig. 1) was estimated to be 0.35% [0.28%; 0.45%].

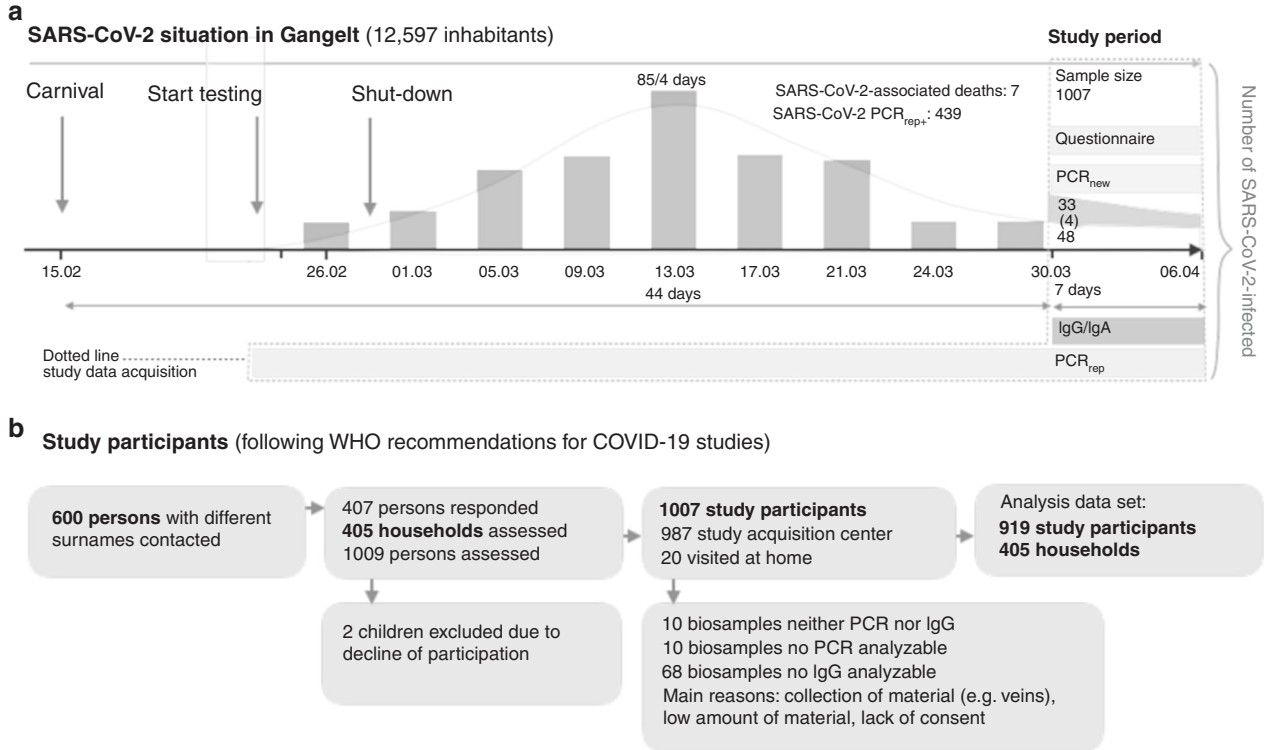

**Fig. 1 Timeline of the SARS-CoV2 super-spreading event and enrollment of study participants. a** On February 15, 2020, a carnival celebration became a SARS-CoV2 super-spreading event in the German community of Gangelt. The resulting increase in SARS-CoV2-infected people was instantly countered by a complete shutdown (schools, restaurants, stores, etc.). As a result, the number of reported cases ($PCR_{rep}$) reached its peak around March 13 (85 reported in a four-day period) and declined thereafter, with 48 new cases reported in the seven-day study period (March 30 to April 6). Thus, at the starting point of the study (March 30), the main wave of new infections had already passed. The number of PCR-positive cases found in the study population ($PCR_{new}$) was 33 (four of those reported PCR positive in the past). This situation in the community of Gangelt was ideal to assess the cumulative real number of SARS-CoV2-infected individuals (area within dotted line: $PCR_{rep}$, $PCR_{new}$ and anti-SARS-CoV2 IgG/A). **b** Enrollment and flow of participants through the study.

While the percentage of previously reported cases as identified by questionnaire in the study population was 2.39% ($PCR_{rep+}$), the percentage of officially reported cases in the community of Gangelt at the end of the study period (April 6) was 3.08% (388/ 12,597). If the corresponding correction factor (3.08%/2.39% = 1.29 [0.87; 2.20]) is applied to the infection rate of 15.53% of the study population (corrected for 99% and 98% specificity in dark and light gray respectively, specificity of IgG ELISA see Methods section), the resulting corrected infection rate is 19.98% [14.13%; 32.00%] (Fig. 3b, third bar from left). Accordingly, the corrected higher infection rate would reduce the IFR to an estimated 0.28% ([0.17%; 0.39%]). (Fig. 3c).

**Infection rate, symptoms, and intensity of disease.** The following symptoms were found to be associated with SARS-CoV2 infection (based on $IgG^+$, $PCR_{rep+}$, $PCR_{new+}$, ranked by odds ratios (OR) with 95% CIs, adjusted for sex and age): loss of smell (OR: 19.06 [8.72; 41.68]; $p < 0.001$), loss of taste (OR: 17.01 [8.49; 34.10]; $p < 0.001$), fever (OR: 4.94 [2.87; 8.50]; $p < 0.001$), sweats and chills (OR: 3.74 [2.31; 6.07]; $p < 0.001$), fatigue (OR: 2.99 [1.97; 4.56]; $p < 0.001$), cough (OR: 2.81 [1.92; 4.11]; $p < 0.001$), muscle and joint ache (OR: 2.42 [1.46; 4.00]; $p = 0.005$), chest tightness (OR: 2.32 [1.31; 4.11]; $p = 0.019$), head ache (OR: 2.28 [1.46; 3.56]; $p = 0.003$), sore throat (OR: 1.92 [1.25; 2.96]; $p = 0.017$), and nasal congestion (OR: 1.91 [1.28; 2.85]; $p = 0.010$), Table 2. The number of symptoms reported by an individual participant served as an indicator for the intensity of the disease and on average was 2.18-fold higher (adjusted for sex and age, 95% CI: [1.78; 2.66]) in SARS-CoV2-infected ($IgG^+$, $PCR_{rep+}$, $PCR_{new+}$) compared to participants without infection (Fig. 4a, $p < 0.001$). In all, 22.22% of infected ($IgG^+$, $PCR_{rep+}$, $PCR_{new+}$) reported no symptoms at all; for the other infected participants symptom numbers varied between 1 and 11 (Fig. 4b). IgG levels of infected study participants were not found to be associated with the number of symptoms (Fig. 4c).

**Association between household size and rate of infection.** The average number of people in household clusters examined in this study was 2.27 (sd = 1.11, range 1–6) compared to the community of Gangelt (2.44 as of 2011), the state of NRW (2.02, as of December 2018) and Germany (1.99, as of December 2018). In the study population, the infection rate was not found to be associated with the number of people in a household cluster (Fig. 5a). In household clusters in which at least one person was infected, the excess per-person infection risk was 17.59%, 18.05%, and 7.11% for 2-, 3-, and 4-person household clusters, respectively (Fig. 5b, black curve compared to lower gray curve). An association between household cluster size and the per-person infection risk was found (Fig. 5b, $p < 0.001$). In this analysis, 15 household clusters with >4 members were omitted due to small numbers. The average percentage of infected persons in these household clusters was 17.33% (0% in 9, 16.66% in 1, 20% in 2, 40% in 1, 80% in 1, 83.33% in 1 household cluster).

**Table 1 Characteristics and co-morbidities of the 919 study participants in 405 households evaluable for infection status.**

| Characteristic | Value |
|---|---|
| Size of household clusters—no. (%) | |
| 1 person | 98 (24.20) |
| 2 persons | 184 (45.43) |
| 3 persons | 59 (14.57) |
| 4 persons | 49 (12.10) |
| 5 persons or more | 15 (3.70) |
| Age | |
| Median (range) | 53 years (1 year–90 years) |
| Distribution—no. (%) | |
| <5 years | 6 (0.65) |
| 5–14 years | 55 (5.98) |
| 15–34 years | 176 (19.15) |
| 35–59 years | 344 (37.43) |
| 60–79 years | 266 (28.94) |
| >79 years | 72 (7.83) |
| Sex—no. (%) | |
| Male | 451 (49.08) |
| Female | 467 (50.82) |
| Diverse | 1 (0.11) |
| IgG—no. (%) | |
| High | 106 (11.53) |
| Intermediate | 19 (2.07) |
| Normal | 794 (86.40) |
| PCR$_{new}$—no. (%) | |
| Positive | 33 (3.59) |
| Negative | 886 (96.41) |
| PCR$_{reported}$—no. (%) | |
| Yes | 117 (12.79) |
| No | 796 (86.99) |
| Not known | 2 (0.22) |
| Missing: 4 | |
| PCR$_{reported}$ positive—no (%) | |
| Yes | 22 (18.97) |
| No | 93 (80.17) |
| Not known | 1 (0.86) |
| Missing: 1 | |
| Lung disease—no (%) | |
| Yes | 107 (11.67) |
| No | 798 (87.02) |
| Not known | 12 (1.31) |
| Missing: 2 | |
| Cardiovascular disease—no (%) | |
| Yes | 128 (13.93) |
| No | 779 (84.77) |
| Not known | 12 (1.31) |
| Missing: 0 | |
| Neurological disease—no (%) | |
| Yes | 41 (4.46) |
| No | 875 (95.21) |
| Not known | 3 (0.33) |
| Missing: 0 | |
| Cancer—no(%) | |
| Yes | 69 (7.52) |
| No | 842 (92.82) |
| Not known | 6 (0.65) |
| Missing: 2 | |
| Diabetes—no (%) | |
| Yes | 79 (8.62) |
| No | 832 (90.73) |
| Not known | 6 (0.65) |
| Missing: 2 | |
| Carnival—no (%) | |
| Yes | 417 (45.52) |
| No | 498 (54.37) |
| Not known | 1 (0.11) |
| Missing: 3 | |

**Associations between sex, age, and co-morbidities with IgA/ IgG, the rate of infection and the number of symptoms.** IgA levels of infected study participants showed a weak positive association with age but were not found to be associated with sex; IgG levels of infected study participants were neither found to be associated with age nor with sex (Supplementary Fig. 2). Neither sex nor age were found to be associated with the rate of infection (Fig. 6a) nor with the severity of infection as indicated by the number of symptoms (Supplementary Fig. 3A) nor with the percentage of asymptomatic cases (not in figure). Neither an increased rate of infection (Fig. 6b) nor a higher number of symptoms (Supplementary Fig. 3B) were found in individuals with co-morbidities. Co-morbidities of infected study participants were not found to be associated with Ig levels (Fig. 7). For infected study participants the self-reported use of medications queried in the questionnaire (ibuprofen, ACE inhibitors or AT1 agonists, not in figure) had no significant associations with the infection rates or number of symptoms.

**Associations between celebrating carnival, rate of infection, and number of symptoms.** Study participants were asked whether they had participated in carnival events. There was a positive association between celebrating carnival and infection (OR = 2.56 [1.67; 3.93], $p < 0.001$, Fig. 8a). Furthermore, there was a positive association between celebrating carnival and the number of symptoms in infected study participants (estimated relative mean increase: 1.63 [1.15; 2.33], $p = 0.007$, Fig. 8b). While the percentage of asymptomatic infected participants was 36% without celebrating carnival, only 16% who had celebrated carnival were asymptomatic (Fig. 8c). Notably, the mean age of the study participants who attended carnival was $41 \pm 20$ years compared to $57 \pm 20$ years for those who have not participated in carnival festivities ($p < 0.001$); furthermore, the mean age of infected study participants was $45 \pm 20$ years for those who attended carnival versus $60 \pm 21$ years who did not attend carnival ($p < 0.001$).

## Discussion

One key parameter to assessing the potential impact that SARS-CoV-2 infection poses on societies is the fatality rate. We set out to determine the infection fatality rate (IFR), which requires an accurate assessment of the number of SARS-CoV-2-infected individuals. The study presented here was performed in the context of the first super-spreading event in Germany and is the first epidemiological study of SARS-CoV2 infection in such a well-defined, high-prevalence community. It revealed that an estimated 15.53% of the population in the community of Gangelt was infected with the virus, which is fivefold higher than the officially reported number of PCR-positives. Based on the estimated percentage of infected people in this population, the IFR was 0.36%. Infection was strongly associated with previously described characteristic symptoms of SARS-CoV2 infection such as loss of smell and taste. The frequency of infection did not significantly differ between age groups and was not found to be associated with sex. Underlying co-morbidities, such as underlying lung disease or cardiovascular disease, did not show significant associations with the rates of infection. Notably, this does not contradict the well-established fact that co-morbidities such as lung disease predispose for severe disease outcomes[5,6]. The use of ACE-inhibiting drugs or ibuprofen did not show significant associations with the infection rates or number of symptoms, as previously speculated[7].

In our study, an infected participant was defined as either PCR positive, anti-SARS-CoV2-IgG positive, or both, thus including present and past infections. As SARS-CoV-2 first appeared in 2020, seropositives were expected to cover all infections except the very recent. This may become different as the pandemic continues, since a decrease in antibody titers over time needs to be considered in the calculation of the IFR. To determine the IFR, the collection of materials and information including the reported cases and deaths was closed at the end of the study acquisition period (April 6), and the IFR was calculated based on those data. However, some of the individuals still may have been acutely infected at the end of the study acquisition period (April 6) and thus may have succumbed to

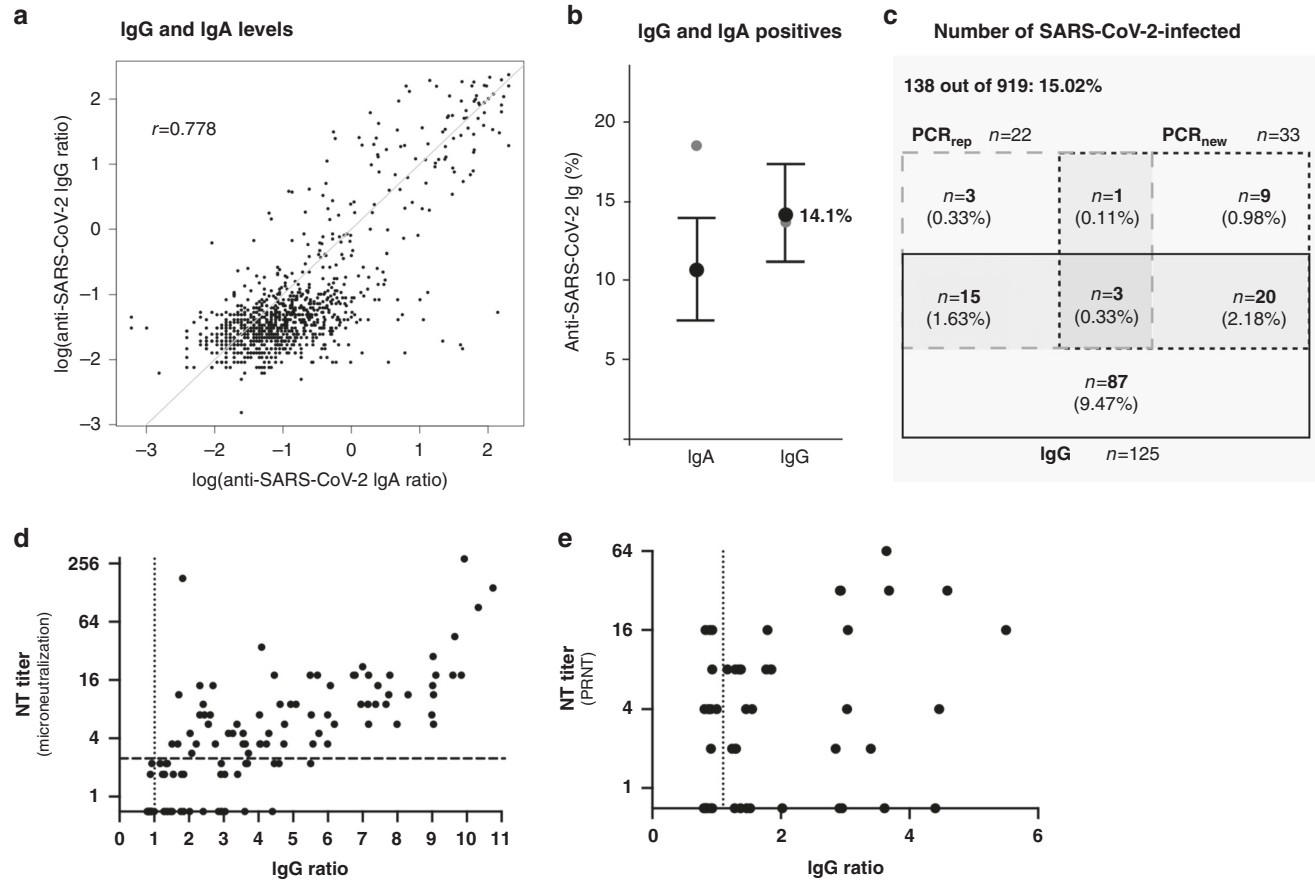

**Fig. 2 IgA and IgG levels and number of SARS-CoV2 infected in the study population.** IgA and IgG were quantified (Euroimmun ELISA) in single plasma samples obtained from the study participants at one time point during the seven-day acquisition period. **a** IgG plotted against IgA in plasma of 919 study participants (log-scale, $r = 0.778$). The gray line indicates equality of log(IgA) and log(IgG). **b** Estimated percentage of IgA reactives (>0.8; black circle: 10.63% [7.48%; 13.88%]; gray circle: raw sample proportion 170/919, 18.50%) and IgG reactives (>0.8; black circle: 14.11% [11.15%; 17.27%]; gray circle: raw sample proportion, 125/919, 13.60%). Estimates were corrected for household clustering (cluster bootstrap) and for sensitivity and specificity (matrix method) of the IgA (sensitivity 100%; specificity 91.2%) and IgG (sensitivity 90.9%; specificity 99.1%) ELISA. Error bars refer to 95% confidence intervals. **c** Absolute numbers of IgG reactives (rectangle with black border), PCRnew positives (rectangle with dashed border, left side) and PCRrep positives (rectangle with dashed border, right side) as well as the respective overlaps of values (percentages in brackets). The number of infected (total gray area) is defined as participants positive for at least one of either IgG, PCRnew, or PCRrep (138/919, 15.02%; raw percentages not corrected for sensitivity and specificity). **d** NT titers were determined by a microneutralization assay using 100 $TCID_{50}$. Titers indicate the reciprocal value of the plasma dilutions that protect 50% of the wells at incubation with 100 $TCID_{50}$. Samples able to suppress the cytopathic effect (CPE) in at least all three wells of the 1:2 dilution (NT titer ≥ 2.8) are depicted above the dashed line. Samples for which the CPE was suppressed in one or two wells of the 1:2 dilution are shown directly below the dashed line. Samples showing a CPE in all wells with either equal or reduced severity compared to the negative control were depicted at the level of the x-axis. **e** Samples below the dashed line in **d** were re-evaluated using a plaque reduction neutralization test (PRNT). The neutralizing titers were calculated as the reciprocal of serum dilutions resulting in neutralization of 50% input virus ($NT_{50}$). Dotted line: upper borderline for ELISA IgG ratio. Source data for **b** is provided as a Source Data file.

the infection later on. In fact, in the 2-week follow-up period (until April 20) one additional COVID-19 associated death was registered. The inclusion of this additional death would bring up the IFR from 0.36% to an estimated 0.41% ([0.33%; 0.52%] in Gangelt, [0.21%; 0.84%] if accounting for uncertainty in the number of deaths). On the other hand, in the same situation including the 8th death, correction for underrepresentation of reported PCR positives would bring the estimated IFR from 0.41% down to 0.32% [0.20%; 0.45%] in the community of Gangelt.

Although the IFR is less variable than the infection rates in different parts of the country, the IFR may still be affected by certain circumstances. The community in which this study was performed experienced a super-spreading event. The IFR was unlikely affected by an overwhelmed healthcare system because sufficient numbers of ICU beds and ventilators were available at all times. However, it is possible that the super-spreading event

itself caused more severe cases. In our study, we found a highly significant increase in both infection rate and number of symptoms when people attended carnival festivities, as compared to people who did not celebrate carnival. This association with carnival was at the same level when adjusted for the age of the participants. Correspondingly, the percentage of asymptomatic individuals was much higher in non-carnival attending infected individuals (more than 35%). At this point, the reason for the association with celebrating carnival remains speculative. However, it is well-established that particle emission and super-emission during human speech increase with voice loudness[8]. Notably, results from experimental human influenza infection studies have demonstrated that the symptom score depends on the viral dose administered[9,10]. Similar observations have been made for MERS[11] and SARS[12,13]. Future studies designed to specifically analyze the infection chains after super-spreading

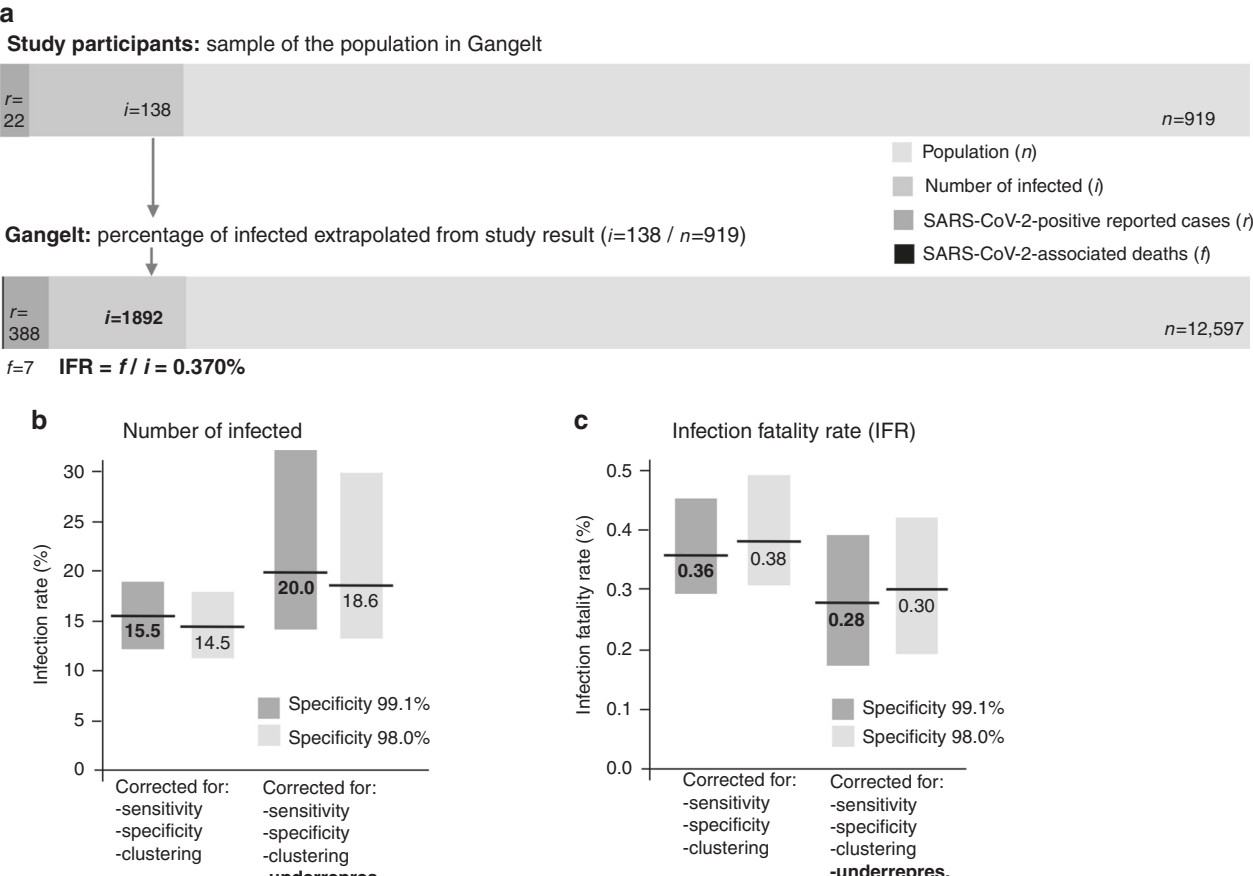

**Fig. 3 Estimation of the SARS-CoV2 infection rate and the IFR. a** The number of SARS-CoV2-positive reported cases in the study population is $r = 22$. The observed number of infected in the study population is known from the data available (at least one of either IgG +, PCR$_{new}$ + or PCR$_{rep}$ +, $i = 138$). The ratio of $i/n$ (study participants, 919) = 0.1502 is a raw estimate of the number of infected in the whole population of Gangelt ($i = 0.1502 \times 12{,}597 \approx$ 1892). A raw estimate of the IFR in Gangelt is given by the number of SARS-CoV2-associated deaths ($f = 7$)/($i = 1892$) = 0.370%. **b** The infection rates estimated from the IgG and PCR data in the study population, corrected for both sensitivity/specificity of the ELISA (matrix method) and household clustering (cluster bootstrap), is 15.53% [12.31%; 18.96%] (left bar, dark gray). An additional correction was made for the underrepresentation of reported PCR positive (PCR$_{rep}$ +) in the study population (22/919 = 0.0239) as compared to the real proportion of PCR$_{rep}$ + in Gangelt (388/12,597 = 0.0308), increasing the infection rate by the factor 0.0308/0.0239 = 1.2866 to 19.98% [14.13%; 32.00%] (third bar from left, dark gray). Note that the latter confidence interval accounts for additional uncertainty in the correction factor. The bars in light gray depict the values corrected for a theoretical specificity of the ELISA of 98% (light gray) instead of the 99% provided on the data sheet of the company. **c** Infection fatality rate calculated based on the estimated infection rate and the number of SARS-CoV2-associated deaths (7 by the end of the acquisition period, mean age 80.8 ± 3.5 years, age range 76 years to 85 years). Similar to the infection rates in **b**, the estimated IFR of 0.36% [0.29%; 0.45%] (left bar) may be an estimate at the upper limit of the real IFR in Gangelt. IFR estimates were obtained by dividing the number of SARS-CoV2-associated deaths (7) by the point estimates and 95% CI limits of the infection rates in **b**. Source data for **b**, **c** are provided as a Source Data file.

events may provide further insight. If substantiated, the IFR under strict hygiene measures might be lower than the IFR in the context of a super-spreading event, as in this study, with important consequences for the strategy against the pandemic. In this context, it is interesting to note that in our study, 22% of all infected individuals were asymptomatic, confirming previous reports[2,3,14]. Notably, asymptomatic infected individuals in our study presented with substantial antibody titers.

In household clusters with at least one infected person we found a relatively moderate excess infection risk, which depended on the household cluster size. Other studies reported a secondary infection risk of 16.3% in China[15] and 7.56% in South Korea[16]. Moreover, comparably low percentages have been seen with other respiratory infections such as influenza (H1N1) 14.5%[17] or SARS 14.9%[18]. Secondary household members may have acquired a level of immunity (e.g., T-cell immunity) that is not detected as positive by our ELISA but still could protect those household members from a manifest infection[9,19].

Whether the IgG levels detected in infected individuals in our study are protective and how long such a protection lasts is currently unknown. Virus neutralization control assays as performed in our study add information but do not provide evidence for the presence of an effective immunity. As with other tests, virus neutralization assays, in general, can be false positive, as cross-reactivity between betacoronaviruses has been reported[20,21]. Likewise a lack of virus neutralization does also not exclude a past infection as there is ample evidence that not all antibody responses neutralize yet may still provide some degree of protective immunity[22,23]. Therefore, at this point our study uses IgG values as an indicator whether an individual was infected but not as evidence for existing immunity. Moreover, a certain degree of protection might exist even if the IgG levels are below the detection threshold of the ELISA. Such individuals are not counted as infected in our study, yet this hidden number of infected could possibly represent an important component towards immunity in a population. The analysis of

**Table 2 Associations between symptoms and infection rate in the 919 study participants.**

CoV-2 infected—no.

| Symptom —no. | | Yes | No | Odds ratio [0.95 CI] | *p*-value[a] | Odds ratio [0.95 CI] | *p*-value[b] |
|---|---|---|---|---|---|---|---|
| Loss of taste | Yes | 37 | 15 | 17.44 [8.71; 34.91] | <0.001 | 17.01 [8.49; 34.10] | <0.001 |
| | No | 100 | 764 | | | | |
| Loss of smell | Yes | 31 | 11 | 19.54 [9.03; 42.30] | <0.001 | 19.06 [8.72; 41.68] | <0.001 |
| | No | 106 | 767 | | | | |
| Fever | Yes | 33 | 47 | 4.63 [2.70; 7.94] | <0.001 | 4.94 [2.87; 8.50] | <0.001 |
| | No | 105 | 732 | | | | |
| Head ache | Yes | 49 | 148 | 2.28 [1.47; 3.54] | 0.002 | 2.28 [1.46; 3.56] | 0.003 |
| | No | 88 | 631 | | | | |
| Cough | Yes | 71 | 205 | 2.77 [1.90; 4.06] | <0.001 | 2.81 [1.92; 4.11] | <0.001 |
| | No | 67 | 575 | | | | |
| Nose congestion | Yes | 59 | 205 | 1.90 [1.27; 2.86] | 0.013 | 1.91 [1.28; 2.85] | 0.010 |
| | No | 75 | 576 | | | | |
| Sore throat | Yes | 42 | 144 | 1.96 [1.27; 3.02] | 0.014 | 1.92 [1.25; 2.96] | 0.017 |
| | No | 93 | 634 | | | | |
| Shortness of breath | Yes | 12 | 34 | 2.08 [1.07; 4.06] | 0.127 | 1.98 [1.01; 3.91] | 0.191 |
| | No | 126 | 741 | | | | |
| Other respiratory symptoms | Yes | 14 | 55 | 1.51 [0.82; 2.78] | 0.556 | 1.49 [0.80; 2.77] | 0.640 |
| | No | 121 | 716 | | | | |
| Fatigue | Yes | 59 | 148 | 3.05 [2.01; 4.63] | <0.001 | 2.99 [1.97; 4.56] | <0.001 |
| | No | 78 | 633 | | | | |
| Sweats and chills | Yes | 40 | 76 | 3.74 [2.31; 6.06] | <0.001 | 3.74 [2.31; 6.07] | <0.001 |
| | No | 97 | 700 | | | | |
| Muscle and joint ache | Yes | 34 | 90 | 2.46 [1.49; 4.08] | 0.004 | 2.42 [1.46; 4.00] | 0.005 |
| | No | 103 | 688 | | | | |
| Stomach pain | Yes | 6 | 45 | 0.64 [0.26; 1.53] | 0.623 | 0.62 [0.26; 1.50] | 0.640 |
| | No | 131 | 735 | | | | |
| Nausea and vomiting | Yes | 9 | 35 | 1.46 [0.50; 4.29] | 0.623 | 1.29 [0.47; 3.56] | 0.640 |
| | No | 128 | 744 | | | | |
| Chest tightness | Yes | 19 | 47 | 2.38 [1.36; 4.16] | 0.014 | 2.32 [1.31; 4.11] | 0.019 |
| | No | 118 | 732 | | | | |

Missing and unknown values in the symptom variables were listwise deleted; *p*-values were corrected for multiplicity using the Bonferroni–Holm procedure. All statistical were two-sided. Adjustments for multiple comparisons were made as indicated.
[a]Corrected for clustering.
[b]Corrected for clustering, age, and sex.

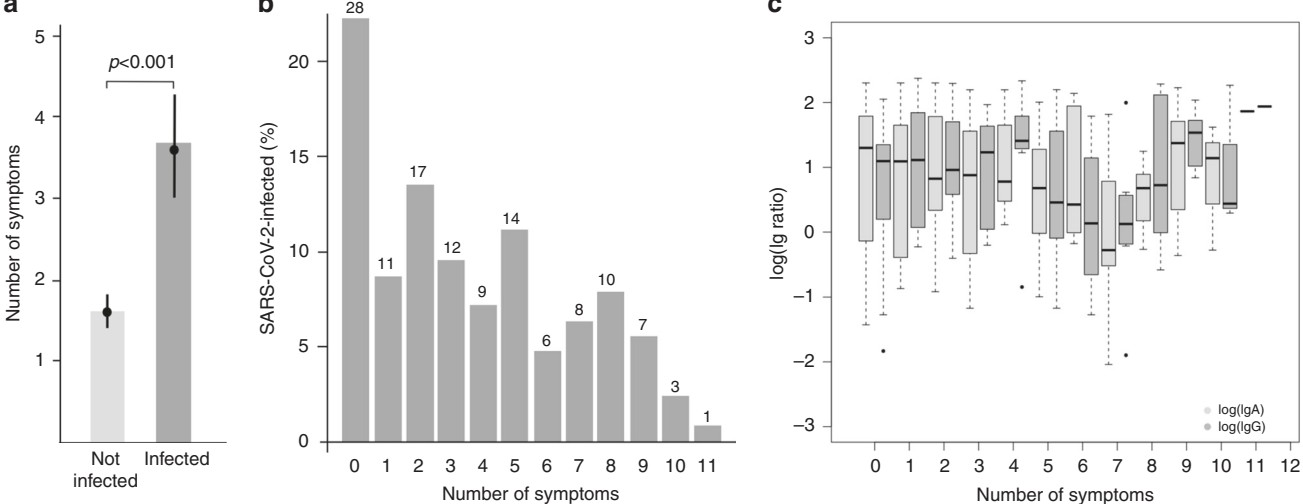

**Fig. 4 Number of symptoms and Ig in SARS-CoV2-infected study participants.** Clinical symptoms reportedly associated with SARS-CoV2-infection were analyzed (questionnaire data). **a** Estimated mean number of symptoms in non-infected study participants (1.61 [1.42; 1.81]) and SARS-CoV2-infected study participants (3.58 [3.01; 4.27], Poisson GEE model, estimated relative mean increase in infected = 2.23 [1.82; 2.73], *p* < 0.001, rho = 0.248 [0.164; 0.332]; Poisson GEE model adjusted for age and sex: estimated relative mean increase in infected = 2.18 [1.78; 2.66], *p* < 0.001, rho = 0.250 [0.167; 0.333]). Results are based on the 876 study participants without missing values in any of the symptom items (range of the observed numbers of symptoms: 0 to 12, mean = 1.92, sd = 2.59, median = 1). Bars refer to the empirical mean values. **b** Raw percentages of symptoms in the 126 SARS-CoV2-infected study participants without missing values for any of the symptoms. Of the SARS-CoV2 infected, 22.22% reported that they did not have any (most left bar on x-axis: 0) of the 15 symptoms. Numbers above bars indicate the total number of individuals in the respective group. **c** IgA and IgG levels and intensity of symptoms. The boxplots depict the log(IgA) (light gray) and log(IgG) (dark gray) levels in the 126 infected study participants. In a quasi-Poisson model, no association between the number of symptoms (response variable) and log(IgA) (covariable) was found. Similar results were obtained from a quasi-Poisson model with the number of symptoms as response variable and log(IgG) as covariable. Note: Quasi-Poisson models were used instead of Poisson GEE models because the number of households was large relative to the number of analyzed study participants. All statistical tests were two-sided. No adjustments for multiple comparisons were made. Source data are provided as a Source Data file.

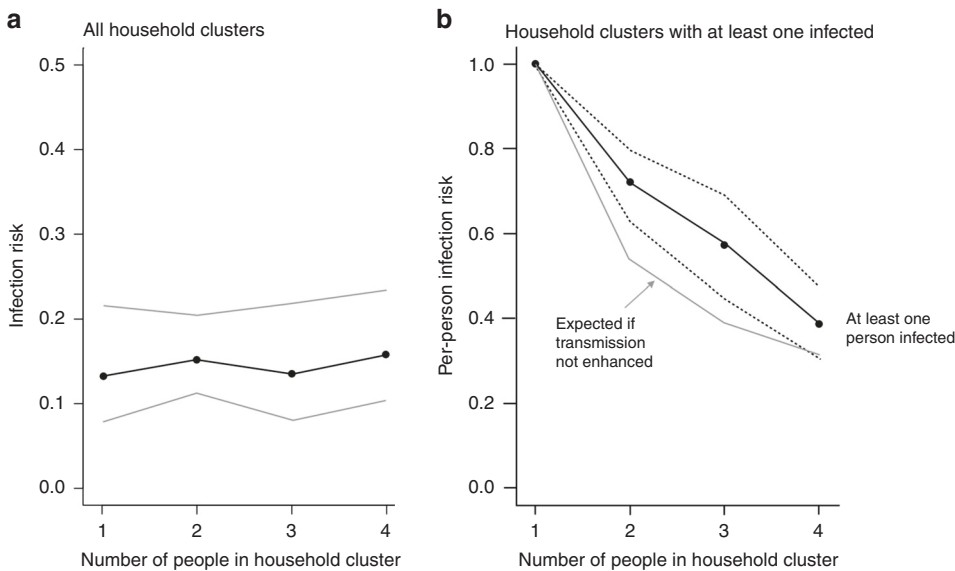

**Fig. 5 Association between household cluster size and the per-person infection risk.** Owing to the sampling procedure, study participants were clustered within households. **a** Estimated per-person infection risk by household cluster size (black dots; 95% CIs: gray lines). Estimates and CI limits were determined by fitting a logistic GEE model with the infection status as response variable and household cluster size as a factor covariable. No association between household cluster size and the per-person infection risk was found ($p = 0.933$). **b** Per-person infection risk in household clusters in which at least one person was found infected (black curve based on 86 household clusters, with 213 persons). The gray line below the black curve shows the expected per-person infection risk under the assumption that infection statuses of the household cluster members are independent. Estimates and CI limits were determined by fitting a logistic GEE model with the infection status as response variable and household cluster size as a factor covariable (excluding 13 household clusters of size 1 each). An association between household cluster size and the per-person infection risk was found ($p < 0.001$). The excess per-person infection risks are given by the deviations of the black curve from the gray reference curve (71.79% − 54.21% = 17.59%, 57.14% − 39.09% = 18.05%, and 38.75% − 31.64% = 7.11%, respectively, for 2-, 3-, and 4-person household clusters). Under the assumption that transmissions to household members occurred independently and were due to only one infected person in each household, the black curve further allows for an estimation of the secondary infection risk: Assigning a 100% risk to the already infected household member, these risk estimates are given by (71.79% × 2 − 100%) = 43.59%, (57.14% × 3 − 100%)/2 = 35.71%, and (38.75% × 4 − 100%)/3 = 18.33%, respectively, for 2-, 3-, and 4-person household clusters (compared to an unconditional estimated infection risk of 15.53%). All statistical tests were two-sided. No adjustments for multiple comparisons were made. Source data are provided as a Source Data file.

anti-SARS-CoV2 IgM might help to further close this window in the future.

It is important to note that the infection rate in Gangelt is not fully representative for other regions in Germany or other countries. Possible limitations of this study include the following: (i) as shown in Supplementary Fig. 1, the age group of 65 years or older is overrepresented in the study cohort compared to the community of Gangelt, the state of NRW and Germany. Reasons could be that elderly people due to retirement may have had more time to participate in a study; second, there may have been a higher awareness of risk because severe cases are presumed to be more likely to occur in this age group; and third, immobile individuals were offered to be visited at home. (ii) Study participants (all and infected) who attended carnival were younger (mean of 41 ± 20 and 45 ± 20 years) than study participants who did not attend carnival (mean of 57 ± 20 and 60 ± 21 years). Therefore, it is possible that due to the younger age of participants in the super-spreading event in Gangelt, the age distribution of the infected persons is different from the age distribution of infected in situations without a super-spreading event, with a possible bias of the IFR towards younger age. In fact, since all individuals who died in the community until the end of the study period were older than 65 years, the present study allows for calculating an estimate of the IFR specifically for the group of older people (>65 years). This estimate is given by 7/(estimated number of infected people in the community ≥ 65 years) = 1.93% [1.39%; 3.05%] and is clearly higher than the respective estimate for the whole population in the community (0.36%). (iii) Study

participants who attended carnival festivities may have been more aware of the risk to acquire an infection and may have been more prone to monitor and recall symptoms. Thus, the possible differences in reporting and recall rates, rather than exposure to a higher viral load, may partially explain the higher number of symptoms in study participants who attended carnival.

Despite the limitations discussed above, the IFR calculated here remains a useful metric for other regions with higher or lower infection rates. In a theoretical model, if the IFR calculated here were applied to Germany, with a number of 6575 SARS-CoV-2-associated deaths (May 2nd, 2020, RKI), the estimated number of infected in Germany would be higher than 1.8 Mio (i.e., 2.2% of the German population). It will be very important to determine the true average IFR for Germany. However, because of the infection rate of ~2% in May 2020 as estimated based on the IFR, an ELISA with 99% specificity will not provide reliable data. Therefore, under the current non-super-spreading conditions, it is more reasonable to determine the IFR in high- prevalence hotspots such as in the community studied here. Consequently, the data of the study reported here will serve as baseline for follow-up studies to identify the corresponding IFR under different hygiene conditions.

## Methods
**Study design, sampling, and procedures**. This study was conducted between March 31st, 2020 and April 6th, 2020 in Gangelt, a community with 12,597 inhabitants (as of January 1st, 2020) located in the German county of Heinsberg in North Rhine-Westphalia. For this cross-sectional epidemiological study, all inhabitants of

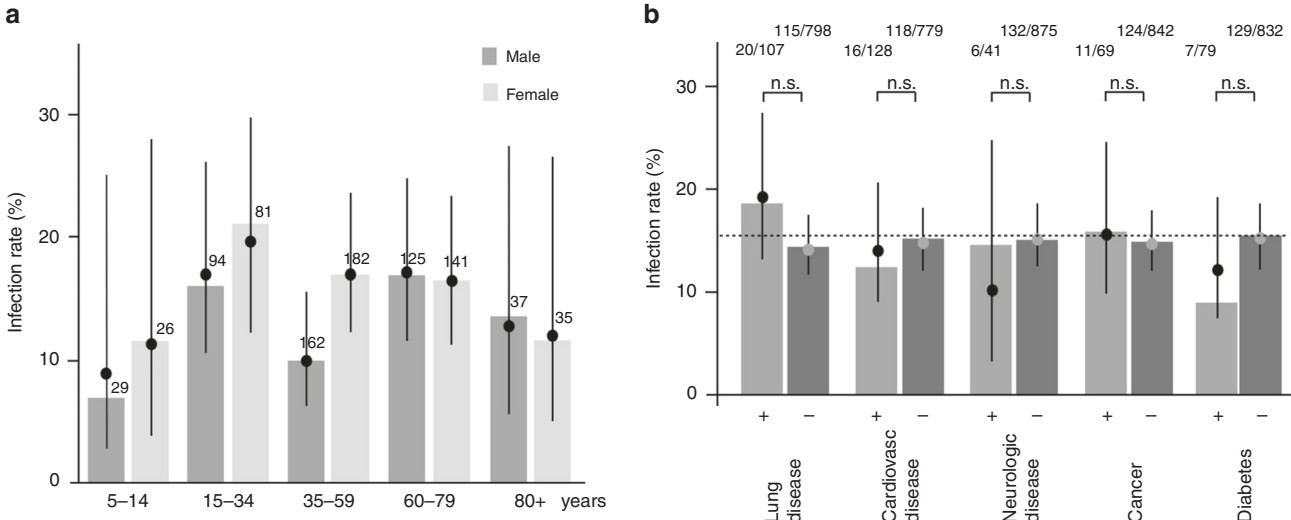

**Fig. 6 Associations of sex, age, and co-morbidities with infection rate. a** Estimated rates of infected in the study participants (filled circles, with 95% CIs) for male participants (dark gray) and female participants (light gray) stratified by age groups. Estimates were obtained by fitting logistic GEE models with the infection status as response variable and age as covariable (rho = 0.256 [0.104; 0.407] and rho = 0.244 [0.154; 0.334], respectively, in the male and female subgroups). Bars refer to the raw percentages. In a logistic GEE model with both sex and age as covariables, neither sex (OR = 1.28 [0.95; 1.73] for females, p = 0.101) nor age (OR = 1.03 [0.94; 1.14] per 10 years, p = 0.539) were found to be associated with infection status. Numbers above bars indicate the total number of individuals in the respective group. Centers of error bars refer to point estimates. **b** For each of the co-morbidities, the infection rate (%) was determined by fitting a logistic GEE model with infection status as response variable to the data of all study participants (light gray: co-morbidity present (+), dark gray: co-morbidity not present (-)). Point estimates obtained from the GEE models are represented by filled circles (with 95% CIs). The bars represent the raw percentages of infected in each of the subgroups (calculated from the participant numbers shown above the bars). No associations between the infection status and any of the co-morbidities were found (according to Bonferroni–Holm corrected p-values). Associations remained insignificant in GEE models that included sex and age as additional covariables. Raw proportions are indicated above bars. Centers of error bars refer to point estimates. Error bars refer to 95% confidence intervals. All statistical tests were two-sided. Adjustments for multiple comparisons were made as indicated. Source data are provided as a Source Data file.

Gangelt were eligible. Enrollment was based on a sample of 600 persons contained in the Heinsberg civil register ("Melderegister"), which is the public authority that collects all names and addresses of the inhabitants of Gangelt. Sampling was done randomly under the side condition that all 600 persons had different surnames, as it was assumed that different surnames were likely to indicate different households. After sampling, the 600 selected persons were contacted by mail and were invited to the study acquisition center, which was established at the site of a public school in Gangelt. The letters sent to the 600 selected persons also included invitations for all persons living in the respective households to participate in the study. Persons aged older than 80 years or immobile were offered the opportunity to be visited at home. For children under 18 years, written and informed consent was provided by the persons with care and custody of the children following aged-adapted participant information. After having written and informed consent, study participants completed a questionnaire querying information, including demographics, symptoms, underlying diseases, medication and participation in carnival festivities (main carnival session "Kappen-sitzung" and others). Furthermore, study participants were asked to provide blood specimens and pharyngeal swabs. In addition to the data provided by the study participants, aggregated data on mortality and socio-demographic characteristics were collected. The latter data were provided by the district administration of Heinsberg and the Statistics & IT Service of the German federal state of North Rhine-Westphalia.

The results presented were obtained in the context of the larger study program termed COVID-19-Case-Cluster-Study. The study was approved by the Ethics Committee of the Medical Faculty of the University of Bonn (approval number 085/20) and has been registered at the German Clinical Trials Register (https://www.drks.de, identification number DRKS00021306, study arm 1). The study was conducted in accordance with good clinical (GCP) and epidemiological practice (GEP) standards and the Declaration of Helsinki (2013), except that, due to time constraints in the situation of the pandemic, this epidemiological non-interventional study was registered April 14, 2020 shortly after the study period (March 31 to April 6, 2020).

Blood was centrifuged and EDTA-plasma was stored until analysis (−80 °C). Analyses were performed in batches at the central laboratory of the University Hospital Bonn (UKB), which is accredited according to DIN EN ISO 15189:2014. Anti-SARS-CoV-2 IgA and Anti-SARS-CoV-2 IgG were determined with enzyme-linked immunosorbent assays (ELISA) on the EUROIMMUN Analyzer I platform (most recent CE version for IgG ELISA as of April 2020, specificity 99.1%, sensitivity 90.9%, data sheet as of April 7th, 2020, validated in cooperation with the Institute of Virology of the Charité in Berlin, and the Erasmus MC in Rotterdam, Euroimmun, Lübeck, Germany). The data sheet (April 7th, 2020)

reports cross-reactivities with anti-SARS-CoV-1-IgG-antibodies, but not with MERS-CoV-, HCoV-229E-, HCoV-NL63-, HCoV-HKU1-, HCoV-OC43-, HCoV-229E-, or HCoV-NL63-IgG antibodies. In our study, infected included positives (ratio of 1.1 or higher, 91% positive in neutralization assay) and equivocal positives (ratio 0.8 to 1.1, 56% positive in neutralization assays). Assays were performed in line with the guidelines of the German Medical Association (RiliBÄK) with stipulated internal and external quality controls. Pharyngeal swabs were stored in Universal Transport Medium Viral Stabilization Media at 4 °C at the study acquisition center for up to 4 h. The cold chain remained uninterrupted during transport. At the Institute of Virology of the UKB swab samples were homogenized by short vortexing, and 300 μl of the media containing sample were transferred to a sterile 1.5 ml microcentrifuge tube and stored at 4 °C. Viral RNA was extracted on the chemagic™ Prime™ instrument platform (Perkin Elmer) using the chemagic Viral 300 assay according to manufacturer's instructions. The RNA was used as template for three real time RT-PCR reactions (SuperScript™III One-Step RT-PCR System with Platinum™ TaqDNA Polymerase, Thermo Fisher) to amplify sequences of the SARS-CoV-2 E gene[24] (primers E_Sarbeco_F1 and R, and probe E_Sarbeco_P11), the RdRP gene (primers RdRP_SARSr_F, and R, and probe RdRP_SARSr-P21), and an internal control for RNA extraction, reverse transcription, and amplification (innuDETECT Internal Control RNA Assay, Analytik Jena #845-ID-0007100). Samples were considered positive for SARS-CoV-2 if amplification occurred in both virus-specific reactions. All PCR protocols and materials were used according to clinical diagnostics standards and guidelines of the Virology Diagnostics Department of the UKB. Neutralization assays were performed using a SARS-CoV-2 strain isolated in Bonn from a throat swab of a patient from Heinsberg. Plasma samples from study participants were inactivated at 56 °C for 30 min. In a first round, neutralizing activity was analyzed by a microneutralization test using 100 TCID50[25]. Serial twofold dilutions (starting dilution 1:2, 50 μl per well) of plasma were performed and mixed with equal volumes of virus solution. All dilutions were made in Dulbecco's modified Eagle medium (Gibco) supplemented with 3% fetal bovine serum (FBS, Gibco) and each plasma dilution was run in triplicate. After incubation for 1 h at 37 °C, $2 \times 10^4$ Vero E6 cells were added to each well and the plates were incubated at 37 °C for 2 days in 5% $CO_2$ before evaluating the cytopathic effect (CPE) via microscopy. In each experiment, plasma from a SARS-CoV-2 IgG-negative person was included and back titration of the virus dilution was performed. Titers were calculated according to the Spearman-Kaerber formula[26] and are presented as the reciprocals of the highest plasma dilution protecting 50% of the wells. To further assess the neutralizing activity of plasma samples exhibiting neutralizing antibody titers

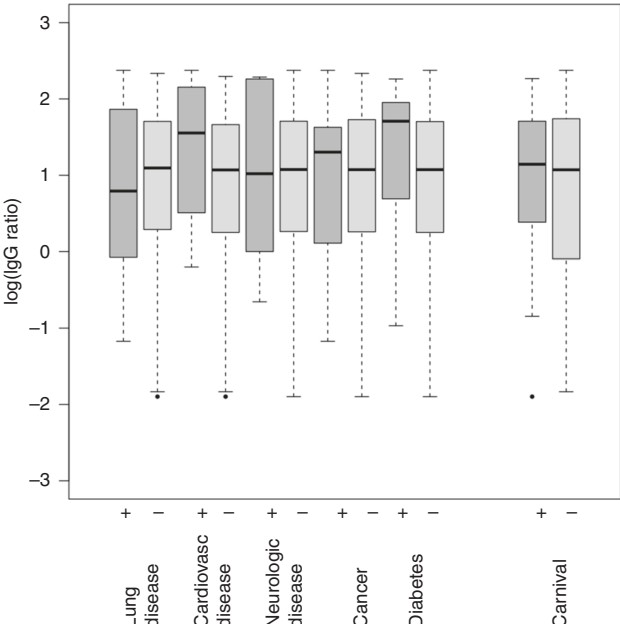

**Fig. 7 Associations between IgG levels, co-morbidities, and celebrating carnival in the infected study participants.** The boxplots depict the log (IgG) levels of the infected study participants in subgroups defined by the presence of co-morbidities and celebrating carnival. In Gaussian models with log(IgG) as response variable, no associations between log(IgG), the co-morbidities, and celebrating carnival were found. Analysis was based on the 127 infected study participants that had complete data in the co-morbidity and carnival variables. Absolute numbers of study participants (left to right): 19, 108, 15, 112, 6, 121, 11, 116, 7, 120, 85, 42. Source data are provided as a Source Data file.

below 2.8 in the microneutralization test, a plaque reduction neutralization test was performed. To this end, heat inactivated plasma samples were serially twofold diluted starting with 1:2 up to 1:1024. One-hundred twenty microliters of each plasma dilution was mixed with 100 plaque forming units (PFU) of SARS-CoV-2 in 120 μl OptiPRO$^{TM}$SFM (Gibco) cell culture medium. After incubation of 1 h at 37 °C, 200 μl of each mixture were added to wells of a 24-well plate seated the day before with $1.5 \times 105$ Vero E6 cells/well. After incubation for 1 h at 37 °C, the inoculum was removed and cells were overlayed with a 1:1 mixture of 1.5% carboxymethylcellulose (Sigma) in 2xMEM (Biochrom) with 4% FBS (Gibco). After incubation at 37 °C for three days in 5% $CO_2$, the overlay was removed and the 24-well plates were fixed using a 6% formaldehyde solution and stained with 1% crystal violet in 20% ethanol.

**Data management and quality control**. Planning and conduct of the study were supported by the Clinical Study Core Unit (Studienzentrale) of the Study Center Bonn (SZB). Support included protocol and informed consent development following specifications of the World Health Organization with regards to pandemic events[5], data management, submission to the ethics committee, clinical trial monitoring and quality control. Study data were collected and managed using REDCap (version 9.5.6) electronic data capture tools hosted at SZB[27,28]. REDCap (Research Electronic Data Capture) is a secure, web-based software platform designed to support data capture for research studies, providing (1) an intuitive interface for validated data capture; (2) audit trails for tracking data manipulation and export procedures; (3) automated export procedures for seamless data downloads to common statistical packages; and (4) procedures for data integration and interoperability with external sources. Questionnaire data were recorded on site using paper case report forms and were entered into the electronic study database using double data entry by trained study personnel. Comparisons between entries were made by the data management unit of the SZB; non-matches were corrected, and duplicated entries were deleted, after assessing the original paper case report forms. Additionally, plausibility checks of demographic data were performed. Study personnel were trained with respect to informed consent and study procedures prior to inclusion of first study participant. The study team was supported on site in Gangelt by a quality control manager who refined workflow processes and monitored critical processes such as obtaining informed consent. Furthermore, regulatory advice could be given whenever asked for or needed. Data entry personnel was trained for double data entry prior to data entry and only then granted database access authorization. Contact with the responsible data managers

could be established when needed. Diagnostic data were imported into the trial database automatically via validated interfaces. Following the completion of the study, critical data was monitored by an experienced clinical trial monitor, which included (but was not limited to) a check of availability of source data (completed questionnaires), random source data verification of diagnostic data and a check of signatures of all informed consent forms obtained.

**Statistical analysis**. In the absence of any pilot data on SARS-CoV-2 infection rate in Gangelt, sample size calculations were based on the WHO population-based age-stratified seroepidemiological investigation protocol for COVID-19 virus infection[29]. According to the recommendations stated in the protocol, a size of 200 samples is sufficient to estimate SARS-CoV-2-prevalence rates <10% with an expected margin of error (defined by the expected width of the 95% confidence interval associated with the seroprevalence point estimate obtained using binomial likelihood) smaller than 10%. In order to rule out larger margins of error due to dependencies of persons living in the same household and to be able to analyze seroprevalence (i.e., infection rates) also in subgroups defined by participant age, it was planned to recruit 1000 participants living in at least 300 households. Statistical analysis was carried out by two independently working statisticians (M.S., M.B.) using version 4.0.0 of the R Language for Statistical Computing (R Core Team 2020: R: A Language and Environment for Statistical Computing, R Foundation for Statistical Computing, Vienna, Austria) and version 9.4 of the SAS System for Windows (copyright © 2002-2012 by SAS Institute Inc., Cary, NC, USA). Participants with a missing anti-SARS-CoV-2 IgG or PCR test result were excluded from analysis, as they were not evaluable for infection status. Participants that did not report a previous positive PCR test result were documented as PCR$_{rep}$ negative. Missing and unknown values in the co-morbidity and symptom variables were not imputed, as listwise deletion reduced sample sizes by less than 5%. Age groups were formed according to the classification system of the Robert Koch Institute (RKI), which is the German federal government agency and research institute responsible for infectious disease control and prevention.

Descriptive analyses included the calculation of means (plus standard deviations, sds) and medians (plus minimum and maximum values) for continuous variables, and numbers (n, with percentages) for categorical variables. Associations between continuous variables were analyzed using the Pearson correlation coefficient (r). Boxplots were generated using the R Language for Statistical Computing.

Generalized estimation equations (GEE)[30] with exchangeable correlation structure within household clusters were used to adjust point estimates and confidence intervals (CIs) for possible dependencies between participants living in the same household. Adjustments for possible sex and age effects were made by including these variables as additional covariables in the GEE models (age in years). One person of diverse sex (Table 1) was excluded from the models including sex as covariable. For binary outcomes (e.g., infection status), GEE models with a logistic link function were applied. Results of logistic GEE models are presented in terms of either back-transformed mean estimates (GEE models with a single covariable) or odds ratios (ORs, GEE models with ≥1 covariables). Note that odds ratio estimates obtained from a GEE model with logistic link function are "population-averaged" in the sense that they represent ratios of population odds but not ratios of an individual's odds. For count data (e.g., number of symptoms), Poisson GEE models with a logarithmic link function were used. Results of Poisson GEE models are presented in terms of either back-transformed mean estimates (GEE models with a single covariable) or estimated relative mean increases/decreases (GEE models with ≥1 covariables). For each GEE model, the estimated correlation between participants living in the same household cluster (rho) is reported. Wald tests were used for hypothesis testing.

All CIs presented in this work were computed using the 95% level. CIs are Wald CIs and were not adjusted for multiple comparisons unless otherwise stated. All statistical hypothesis tests were two-sided. The Bonferroni–Holm procedure was applied to adjust p-values for multiple comparisons as indicated.

Infection rates obtained from IgG and IgA measurements were additionally corrected for possible misclassification bias using the matrix method[31], with sensitivity and specificity values obtained from the ELISA manufacturer's (Euroimmun, Lübeck, Germany) validation data sheet (version: April 7th, 2020). No adjustments were made for age and sex, as these variables were not found to be associated with infection status (Fig. 6a). To account for possible clustering effects due to participants living in the same household, confidence intervals for the corrected infection rate estimates were computed using a cluster bootstrap procedure with 10,000 bootstrap samples[32]. With this procedure, household clusters were sampled with replacement. Within sampled clusters, no additional resampling of household members was carried out. The distributions of the bootstrapped corrected infection rate estimates were symmetrical and close to normality (as indicated by normal quantile-quantile plots, sd = 0.01697), and the percentile method was applied to calculate CI limits. CI limits for the IFR were calculated by dividing the number of deaths (7) by the CI limits of the estimated number of infected. Here, the number 7 was considered fixed, as it corresponded to all recorded SARS-CoV2-associated deaths in Gangelt by the end of the study period and was, therefore, not subject to sampling error. In addition, we computed a Bayesian credibility interval for the IFR that accounted for possible uncertainty in the number of SARS-CoV-2-associated deaths. This CI was defined by the empirical 2.5% and 97.5% quantiles of 100,000 samples drawn from a beta distribution with parameters (7 + 1) and (estimated number of infected—7 + 1), where in each of the 100,000 samples the estimated number of infected were sampled from a normal distribution with mean 0.1553 and standard deviation 0.01697 (multiplied by 12,597). This approach was

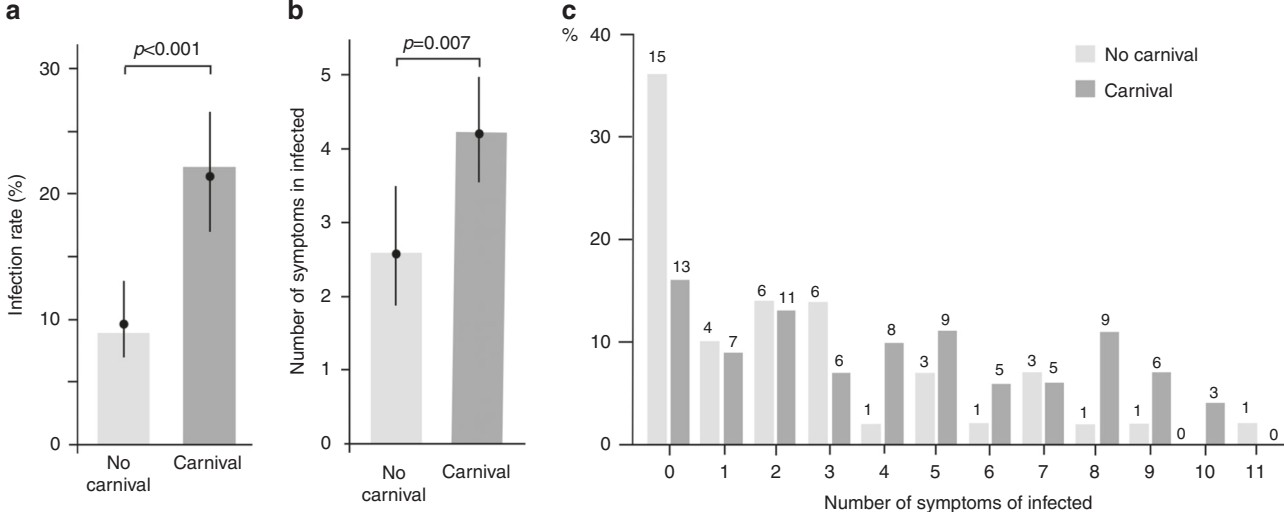

**Fig. 8 Associations of super-spreading event with infection rate and symptoms. a** The association with the lifestyle factor "celebrating carnival" was analyzed (questionnaire "have you celebrated carnival?" yes/no). Celebrating carnival was not limited to attending the main carnival event (Kappensitzung in Gangelt). Estimated infection rate (%; with 95% CIs) of participants not celebrating carnival (light gray) and participants celebrating carnival (dark gray). Point estimates (filled circles) and CIs were obtained by fitting a logistic GEE model with infection status as response variable and carnival (yes/no) as a factor covariable. The bars represent the raw percentage values. There was a positive association between celebrating carnival and infection status (OR = 2.56 [1.67; 3.93], $p < 0.001$, rho = 0.351 [0.162; 0.540]). Similar results were obtained when adding sex and age as covariables to the GEE model (OR = 3.08 [1.92; 4.95], $p < 0.001$, rho = 0.340 [0.126; 0.554]). Analyses were based on the 915 participants that had complete data in both the carnival and the infection variables. Error bars refer to 95% confidence intervals. **b** Estimated mean number of symptoms in infected participants not celebrating carnival (light gray) and in infected participants celebrating carnival (dark gray). Point estimates (filled circles) and CIs were obtained by fitting a quasi-Poisson model with the number of symptoms as response variable and carnival (yes/no) as a factor covariable. The quasi-Poisson model was used instead of a Poisson GEE model because the number of households was large relative to the number of analyzed study participants. There was a positive association between celebrating carnival and the number of symptoms (estimated relative mean increase = 1.63 [1.15; 2.33], $p = 0.007$). Similar results were obtained when adding sex and age as covariables to the model (estimated relative mean increase = 1.62 [1.12; 2.34], $p = 0.011$). Analyses were based on the 124 infected participants that had complete data in both the carnival and infection variables. Error bars refer to 95% confidence intervals. **c** Raw percentages of infected participants celebrating carnival, grouped by their numbers of symptoms. Numbers above bars indicate the total number of individuals in the respective group. All statistical tests were two-sided. Adjustments for multiple comparisons were made as indicated. Source data are provided as a Source Data file.

motivated by the fact that the beta distribution with parameters $(7 + 1)$ and (estimated number of infected—$7 + 1$) is the posterior distribution obtained from a uniform prior distribution on the IFR and a binomial likelihood with estimated number of infected trials and 7 successes. Furthermore, an age-standardized estimate of the IFR in Gangelt was computed. This was done by determining infection rates from the study data in each of the age groups defined in Supplementary Fig. 1 (again corrected for possible misclassification bias using the matrix method) and by calculating an age-standardized estimate of the number of infected in Gangelt (using the proportions of the age groups in Gangelt presented in Supplementary Fig. 1; confidence intervals computed using a cluster bootstrap procedure with 10,000 samples). The age-standardized IFR estimate and its CI limits were calculated by dividing the number of deaths (7) by the age-standardized estimated number of infected and its CI limits, respectively.

For the analysis of household clusters in which at least one person was found infected (Fig. 5b), the expected per-person infection probability under the assumption that infection statuses of the household cluster members are independent (gray line) was computed by evaluating the conditional probability $p$ (person is infected | at least one person in the same household cluster is infected). Assuming the unconditional infection probability to be $p = 0.1553$, the aforementioned conditional probability is derived as $p/(1 − (1 − p)^2) = 0.542$, $p/(1 − (1 − p)^3) = 0.391$, and $p/(1 − (1 − p)^4) = 0.316$ for 2-, 3-, and 4-person household clusters, respectively.

Note: Throughout the paper, the term rate refers to the number of persons experiencing an event divided by the number of the reference population, in line with the definition of the IFR[4]. We adopted this definition due to its widespread use in the context of COVID-19 research, keeping in mind that "rates" are usually defined in terms of person-time (e.g., Rothman et al.[33]).

**Reporting summary**. Further information on research design is available in the Nature Research Reporting Summary linked to this article.

## Data availability
The data contain information that could compromise the privacy of research participants. Data sharing restrictions imposed by national and trans-national data protection laws prohibit general sharing of data. However, upon submission of a proposal to the principal investigator of this study and approval of this proposal by (i) the principal investigator, (ii) the Ethics Committee of the University of Bonn and (iii) the data protection officer of the University Hospital Bonn, data collected for the study can be made available to other researchers. A source data file that contains the numbers presented in the figures (i.e., means, standard deviations etc.) is provided with this paper.

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

## Acknowledgements

The government of the German Federal State of North Rhine-Westphalia unconditionally provided 65,000 Euro to support the study. H.S., E.B., M.N., and G.H. are members of the Excellence Cluster ImmunoSensation EXC 2151—390873048. We would like to thank the inhabitants of Gangelt for their participation. We would also like to thank the local government of Kreis Heinsberg for their support to conduct the study. We thank Sammy Bedoui, Art Krieg and André Karch for critical reading of an early version of the manuscript, and Helmut Küchenhoff for statistical advice. Furthermore, we would like to thank the following people who helped with the study: Gero Wilbring, Janett Wieseler, Marek Korencak, Ryan Nattrass, Jernej Pusnik, Maximilian Becker, Ann-Sophie Boucher, Marc Alexander de Boer, Rebekka Dix, Sara Dohmen, Kim Friele, Benedikt Gansen, Jannik Geier, Marie Gronemeyer, Sarah Hundertmark, Nora Jansen, Michael Jost, Louisa Khorsandian, Simon Krzycki, Ekaterina Kuskova, Judith Langen, Silvia Letmathe, Ann-Kathrin Lippe, Jonathan Meinke, Freya Merker, Annika Modemann, Janine Petras, Sophie Marie Porath, Anna Quast, Laurine Reese, Isabel Maria Rehbach, Jonas Richter, Thea Rödig, Eva Schmitz, Tobias Schremmer, Louisa Sommer, Jennifer Speda, Yuhe Tang, Oliver Thanscheidt, Franz Thiele, Johanna Thiele, Julia Tholen, Sophia Tietjen, Moritz Transier, Maike van der Hoek, Tillmann Verbeek, Sophia Verspohl, Kira Vordermark, Julian Wirtz, Marina Wirtz, Lisa Zimmer, Philip Koenemann, Adi Yaser, Lisa Anna, Katharina Bartenschlager, Lisa Baum, Roxana Böhmer Romero, Diana de Braganca, Isabelle Engels, Moritz Färber, Carina Fernandez Gonzalez, Lucia Maria Goßner, Victoria Handschuch, Franziska Georgia Liermann, Steffan Meißner, Laura Racenski, Patrick, Denis Raguse, Larissa Reiß, Maximilian Rölle, Franziska Scheele, Chiara Schwippert, Arlene Christin Schwippert, Antonia Seifert, Joshua David Stockhausen, Sofia Waldorf, Leonie Weinhold, Nicolai Trimpop, Julia Reinhardt, Vera Gast, Michelle Yong, Eva Engels, Jonathan Meinke, Susanne Schmidt, Janine Schulte, Saskia Schmitz, Kübra Bayrak, Regina Frizler, Katarzyna Andryka, Sofia Soler, Bastian Putschli, Thomas Zillinger, Marcel Renn, Patrick Müller, Dillon Corvino, Zeinab Abdullah, Katrin Paeschke, Hiroki Kato, Florian Schmidt, Daniel Hinze, Martina Schmidt, Arcangelo Ricchiuto, Sonja Gross, Uta Wolber, Marion Zerlett, Esther Sib, Benjamin Marx, Souhaib Aldabbagh. We thank the Medizinischer Dienst der Krankenkasse (MDK) Nordrhein for their generous help in conducting this study. In particular we would like to thank: Tanja Bell, Ina de Hesselle-Taddey, Susanne Goll, Katja Hoffmann-Pruss, Niko Kalamakidis, Verena Kayser, Hildegard Kessler, Norbert Körfer, Daniela Kroll, Florian Messerschmidt, Svenja Peters, Sebastian Schröder, Linda Schroers, Michael Zimmermann, Klaus-Peter Thiele. We thank Stefan Holdenrieder, Alexander Semaan, Bernd Pötzsch, and Georg Nickenig for providing control samples. The idea, the plan, the concept, protocol, the conduct, the data analysis, and the writing of the manuscript of this study were independent of any third parties, including the government of North Rhine-Westphalia, Germany.

## Author contributions

H.S., C.F., E.B., M.C., M.E., R.M.S., M.S., and G.H. contributed to the conception, design, and interpretation of the work. B.S., B.M.K., E.R., T.H., R.D.P., L.W., M.E.B., A.K., A.S., P.H., B.S.W., R.M.S., H.S., M.N., and A.M.E.H. contributed to the data acquisition and analysis of the data. M.N. and A.M.E.H. revised the work critically for important intellectual content. H.S., E.B., M.S., M.B., and G.H. wrote the manuscript.

## Funding

## Competing interests

The authors declare no competing interests.
