## [Peer Review File · Nature Communications]

Reviewers' Comments:

Reviewer #1:

Remarks to the Author:

Comments on NCOMMS-20-22462

Infection fatality rate of SARS-CoV2 in the first super-spreading event in Germany

by Streeck et al

The manuscript describes a well-conducted prospective study on the IFR in a COVID-19 cluster in Germany that deserves to be published. Congratulations to the authors for this thorough investigation with double data entry and two independent statistical analyses. I cannot comment on the technical and virological parts of the paper but will discuss some methodological, epidemiological and statistical aspects.

General comments:

One aspect I was missing in this paper was an investigation of the severity of disease (number of symptoms) and percentage of asymptomatic cases as a function of age. I understand that this is not central to the objective of the paper, but it would be still very interesting to investigate this in more detail.

Another question is whether the estimated IFR is representative for other populations. The authors state on page 9 that "The comparison of age groups in the study population to the community of Gangelt, to the state of North Rhine-Westphalia (NRW), and to Germany is illustrated in Fig. S1", but no interpretation is given. Why is there such a large difference between "Gangelt" (~30%) and "Study" (~20%) for age group 65 yrs and older? Why is the age categorization in this Figure different from the one in Table S1 and S2? Some discussion seems needed and perhaps confidence intervals should be added to the proportions for "Study". Is it possible to age-standardize the IFR?

Sample size calculations (page 4): "a sample of 600 persons aged older than 18 years was drawn from the civil register." On page 7 it says "it was planned to recruit 1,000 participants". This does not seem to match the target sample size of 1400 persons mentioned in the DRKS trial description. Why has the sample size been reduced?

The second paragraph on page 10 describes adjustments for "under-representation". It is based on the percentage 3.08% (388/12,597) of officially reported cases. However, the uncertainty is not taken into account. My own calculations give a 95% Wilson CI from 2.79% to 3.40% which translates to a 95% CI from 1.17 to 1.42 for the correction factor. Should this not be incorporated in the estimates shown in Figure 3C and cited in the text? Likewise, should the uncertainty in PCR_{rep+} not also be incorporated? Related to that, I was wondering why no IFR adjusted for "under-representation" is reported in the abstract nor in the discussion (when taking into account the 8th death).

The investigators are to be congratulated to have two competent statisticians on board, who carried out the analysis even independently. I have a few questions on the exact procedures: Did one perform the analysis with R and the other one with SAS? Was there any discrepancy in the results? Was there a statistical analysis plan (SAP), written before the analysis has been conducted?

Data availability: I couldn't find the data although the Nature reporting summary states that "all data generated or analysed during this study are included in this published article (and its supplementary information files)". I was pleased to see that the authors plan to share individual patient data with other researchers, as already stated in the DRKS trial registration. I would like to encourage the authors to make not only data but also analysis code available in order to follow current Open Science standards.

Minor comments:

Abstract: The last sentence describes a non-monotonic excess risk which is hard to interpret. Perhaps drop or add confidence intervals?

Page 3, bottom: The reader might want to know more about the index case and from where the infection was brought into Gangelt.

I was surprised that the DRKS registration date (2020/04/14) was after the study period (March 30 to April 6, page 8). More details on the exact timing of protocol writing, ethics approval, study conduct, etc seem appropriate.

Page 6: "with regards to events⁴", should this refer to the WHO reference [5]?

page 8: I would suggest to drop the sentence "By definition, GEE models employ quasi-likelihood methods to obtain point estimates and CIs."

page 8, "Statistical analysis was carried out by two independently working statisticians (MS, MB) (further details see suppl. methods)": Was there a statistical analysis plan (SAP)? If yes, then this could be mentioned here.

page 8, bottom: The reader might want to know the age of the 7 deaths (and of the 8th death mentioned in the discussion).

page 10 and following: I was wondering which age categorizations have been used for age-adjustments, there are different ones in Supp Mat (see my comment above).

Perhaps some comments might be appropriate in the Supp Mat that the odds ratio estimates from a logistic regression GEE are "population-averaged", so are ratios of population odds, not ratios of an individual's odds.

I was pleased to see the Rogan and Gladen (1978) approach being used to adjust for misclassification. Is the uncertainty in sensitivity and specificity small enough to be ignored? Can you rule out that misclassification was differential? See e.g.

K. J. Rothman, S. Greenland, and T. L. Lash. Validity in epidemiological studies. In K. J. Rothman, S. Greenland, and T. L. Lash, editors, *Modern Epidemiology*, pages 128–147. Lippincott Williams & Wilkins, Philadelphia, 3rd edition, 2008.

Leonhard Held
15 June 2020

Reviewer #2:

Remarks to the Author:

In this paper, authors perform a detailed investigation of an outbreak of SARS-CoV-2 in a small town in Germany, with a large super-spreading event that occurred during carnival festivities. A large serosurvey was performed to identify infected individuals along with results of PCR testing. Authors estimate that 15% of the population was infected, leading to an IFR of 0.36%. Participation in the carnival increased both the infection rate and the number of symptoms in the infected. Transmission in households was also quantified.

This is a well-conducted and interesting study that provides important insights on the epidemiology of SARS-CoV-2. I have a couple of comments.

1) Dependency of the IFR estimate on the age of infected individuals: Since the severity of SARS-CoV-2 infection drastically increases with age, the estimate of the IFR of SARS-CoV-2 in a population is highly dependent on the proportion of elderly individuals among infected people. With a slight increase of that proportion, one could expect substantial increase in the population-level IFR. It appears that this outbreak was largely driven by a super-spreading event that occurred during carnival festivities. If younger members of the community were more likely to attend this even than elderly people, it is possible that infected people in this outbreak were younger than what they typically are in Germany. As a result, the IFR in this outbreak may be lower than what it would typically be in Germany. Authors need to discuss the dependency of the IFR estimate on the age distribution of infected individuals and how it may potentially lead to IFR in this community being lower than what would typically be expected in Germany. Authors should document the age distribution of those that attended the carnival and compare it to the age distribution of the population. Was age an important risk factor for carnival attendance? Do authors know the ages of individuals who died and could they provide age-specific estimates of IFR?

2) Authors find that infected individuals who attended carnival festivities reported a larger number of symptoms. While this could indeed be due to a dose-response phenomenon, this might also potentially be explained by different reporting/recall rates. Indeed, if the carnival was quickly linked in the media to the outbreak, it is possible that people who attended the carnival were more quickly aware they were at risk and more prone to monitor and recall their symptoms. This possibility should be discussed.

Minor comments:

Page 3: "The current estimate of the CFR in Germany by WHO is between 2.2% and 3.4%". I think this may be a bit confusing for an audience of non-specialists. The CFR is often defined by the proportion of individuals with symptomatic infections who die; and no one believes this proportion is as high as 2-3%. It would be good if authors could provide the more narrow definition they use here and clarify the difference with the proportion of symptomatic infections who die upfront.

NCOMMS-20-22462 Revised manuscript:

Streeck et al, "Infection fatality rate of SARS-CoV2 in the first super-spreading event in Germany"

Point-to-point reply

Reviewer #1

Reviewer 1: The manuscript describes a well-conducted prospective study on the IFR in a COVID-19 cluster in Germany that deserves to be published. Congratulations to the authors for this thorough investigation with double data entry and two independent statistical analyses. I cannot comment on the technical and virological parts of the paper but will discuss some methodological, epidemiological and statistical aspects.

One aspect I was missing in this paper was an investigation of the severity of disease (number of symptoms) and percentage of asymptomatic cases as a function of age. I understand that this is not central to the objective of the paper, but it would be still very interesting to investigate this in more detail.

Reply: We agree that this information is very interesting. Severity of disease (number of symptoms) as a function of age has been already provided in the supplement (Figure S5 A of the manuscript). The percentage of asymptomatic cases as a function of age is provided in the figure below for the information of the reviewer. This figure is not included in the manuscript, but the result as such has now been added to the text of the results section (see revised version of the manuscript, page 12, lines 20-21).

Figure for the reviewers: Associations of sex and age with the percentage of asymptomatic study participants among the infected. Bars refer to the raw percentages of asymptomatic cases in the infected (i.e., to infected study participants not having reported any of the 15 symptoms). Numbers above bars indicate the total numbers of asymptomatic and infected individuals in the respective

group. Note: Due to zero counts and small numbers in some of the groups, no statistical model was fitted to the data.

Furthermore, please note that we revised the legend of Figure S4 (associations between Ig levels, age and sex).

Reviewer 1: Another question is whether the estimated IFR is representative for other populations. The authors state on page 9 that "The comparison of age groups in the study population to the community of Gangelt, to the state of North Rhine-Westphalia (NRW), and to Germany is illustrated in Fig. S1", but no interpretation is given. Why is there such a large difference between "Gangelt" (~30%) and "Study" (~20%) for age group 65 yrs and older? Why is the age categorization in this Figure different from the one in Table S1 and S2? Some discussion seems needed and perhaps confidence intervals should be added to the proportions for "Study". Is it possible to age-standardize the IFR?

Reply: We agree with the reviewer that it is important to discuss these differences in the manuscript. Please note that according to Figure S1, the age group 65 years or older in "Gangelt" is approx. 20 % (black bar), and in the "Study" it is approx. 30 % (grey bar). Thus, older people are overrepresented in the study compared to Gangelt, NRW and Germany. Reasons for overrepresentation of the age group 65 years and older could be: i) more time to participate in the study (e.g., retired), ii) higher awareness of risk because severe cases are more likely to occur in this age group, iii) immobile individuals were offered the opportunity to be visited at home. We now discuss this issue in the new paragraph about limitations of the study on page 16 lines 4-7).

Concerning the question regarding the different age categorizations: for Table S1 and Table S2, we chose the age categorization used by the Robert Koch Institute (federal institute for disease control and prevention in Germany). However, this age categorization was not available for the population data of Gangelt, NRW and Germany, and we had to use the age categories in use by the Landesdatenbank NRW and statista.com which are shown in Figure S1. We therefore have now included this information in the figure legend of Figure S1.

As for the confidence intervals regarding proportions for "Study" and age-standardized IFR: Instead of adding confidence intervals to Figure S1, which represents purely descriptive numbers, we followed the reviewer's suggestion to age-standardize the IFR. For this we calculated infection rates in the age groups of Figure S1 and calculated an age-standardized estimate of the number of infected for Gangelt (using the proportions of the age groups in Gangelt presented in Figure S1). Dividing the number of deaths by the age-standardized number of infected results in an age-standardized IFR estimate of 0.35% [0.28%; 0.45%]. We added this age-standardized IFR estimate to the manuscript (please see abstract page 3, lines 15-16; and page 11, lines 8-9).

Reviewer 1: Sample size calculations (page 4): "a sample of 600 persons aged older than 18 years was drawn from the civil register." On page 7 it says "it was planned to recruit 1,000 participants". This does not seem to match the target sample size of 1400 persons mentioned in the DRKS trial description. Why has the sample size been reduced?

Reply: We agree with the reviewer that this is an important point that needs to be clarified. The Covid-19 Case-Cluster-Study registered as DRKS00021306 consists of two study arms. The present manuscript exclusively reports the results of study arm 1 which was planned to recruit 1,000

participants (IFR and proportion of infected in Gangelt). Given the average household size in this region is ~ 2.3 , we estimated to reach 1000 participants by contacting 600 individuals as index persons. Index persons were asked to bring all household members to the study site. Study arm 2 examines the specific situation of the Kappensitzung (carnival super-spreading event) with the intention of recruiting all participants of the Kappensitzung (approximately 400 participants). The two studies were performed separately (note that the random selection of households in study arm 1 also recruited a few individuals who attended the Kappensitzung). The IFR study is based on a random cohort as described in the manuscript. The target sample size of this main study (1000 participants) has been reached. Study arm 2 (carnival study) intended to include participants of the carnival event. To clarify these issues, we revised the DRKS trial entry accordingly and added information to the methods section where DRKS00021306 is mentioned (Supplementary methods, Study design page 3, lines 18-22).

Reviewer 1: The second paragraph on page 10 describes adjustments for "under-representation". It is based on the percentage 3.08% (388/12,597) of officially reported cases. However, the uncertainty is not taken into account. My own calculations give a 95% Wilson CI from 2.79% to 3.40% which translates to a 95% CI from 1.17 to 1.42 for the correction factor. Should this not be incorporated in the estimates shown in Figure 3C and cited in the text? Likewise, should the uncertainty in PCR_{rep+} not also be incorporated? Related to that, I was wondering why no IFR adjusted for "under-representation" is reported in the abstract nor in the discussion (when taking into account the 8th death).

Reply: Following the suggestion of the reviewer, we adjusted the confidence intervals in Figures 3 B and C by accounting for the uncertainty in the correction factor. Analogous to the confidence intervals for the infection rate & IFR in the left-most pairs of boxplots in Figures 3 B and C (not corrected for under-representation), we computed these confidence intervals using a cluster bootstrap approach. Specifically, in each bootstrap sample we re-calculated the value of the correction factor by dividing the number of reported PCR positives in the bootstrap sample by the respective size of the bootstrap sample, and by subsequently dividing the population fraction 388/12,597 (not subject to sampling error) by the resulting bootstrap sample fraction. We followed the same strategy to calculate the IFR adjusted for under-representation when taking into account the 8th death, which is now reported on page 14 lines 14-16 of the revised manuscript.

Reviewer 1: The investigators are to be congratulated to have two competent statisticians on board, who carried out the analysis even independently. I have a few questions on the exact procedures: Did one perform the analysis with R and the other one with SAS? Was there any discrepancy in the results? Was there a statistical analysis plan (SAP), written before the analysis has been conducted?

Reply: Yes, one statistician (MS) performed the analyses in R, and the other statistician (MB) performed the analyses in SAS. There were no discrepancies in the results apart from very few and very minor differences, which were likely due to numerical issues regarding the implementations of the GEE models in SAS and R (iteration numbers, convergence criteria, etc.) and to different seeds used for the generation of bootstrap samples. Results of R analyses were reported if differences were observed. Also, small parts of the analyses were performed solely in R (e.g. the Bayesian credibility interval for the IFR) but not in SAS. This is apparent in the R and SAS code files, please see attachments 1 and 2.

Due to the time constraints of this specific situation, there was no SAP written before the analysis was conducted. However, we were very clear about the structure and content of the analyses right from the start of the project. Furthermore, we did not make any data-dependent choices for the definitions of any of the involved variables; in particular, all available symptom variables were used for the definition of disease severity. Likewise, we based the categorization of age on an external (non-data-dependent) categorization scheme (RKI, see above).

Reviewer 1: Data availability: I couldn't find the data although the Nature reporting summary states that "all data generated or analysed during this study are included in this published article (and its supplementary information files)". I was pleased to see that the authors plan to share individual patient data with other researchers, as already stated in the DRKS trial registration. I would like to encourage the authors to make not only data but also analysis code available in order to follow current Open Science standards.

Reply: Following the suggestion of the reviewer, we added the R and SAS code as an attachment to this point-by-point reply. When originally registering the trial, we were planning to share individual patient data with other researchers, see DRKS trial registration. However, in the meantime, we were advised by the data protection officer of the University Hospital Bonn to neither publish nor share individual patient data. This is because (i) the study population was only ~12,000 participants, which imposes an extremely high re-identification risk on the study participants, making anonymization of individual data impossible, and (ii) because it turned out that the consent form used for this study does not allow for secondary use of the individual patient data, making online publication of the individual patient data legally impossible.

We have now included this information in the Nature reporting summary: "When originally registering the trial, we were planning to share individual patient data with other researchers as indicated in the DRKS trial registration. However, in the meantime, we have been advised by the data protection officer of the University Hospital Bonn to neither publish nor share individual patient data. This is because the consent form used for this study does not allow for secondary use of the individual patient data, making online publication of the individual patient data impossible.

Minor comments:

Reviewer 1: Abstract: The last sentence describes a non-monotonic excess risk which is hard to interpret. Perhaps drop or add confidence intervals?

Reply: Following the suggestion of the reviewer, we dropped the numbers referring to the excess risk from the abstract. Furthermore, again following the comment of the reviewer, we tried to explain the interpretation of the excess risk in more detail in the legend of Figure 4, also making reference to the term "secondary infection risk" used in the medRxiv version of the paper. We hope that the inclusion of these additional numbers is acceptable in order to better explain the meaning of Figure 4.

Reviewer 1: Page 3, bottom: The reader might want to know more about the index case and from where the infection was brought into Gangelt.

Reply: The identification of the index case is one of the aims of the other arm of the Covid-19 Case-Cluster-Study (Kappensitzung) under the same entry DRKS00021306, see above. This other study arm

aims to track the spread of the infection specifically in the context of the Kappensitzung, and the results will be published separately.

Reviewer 1: I was surprised that the DRKS registration date (2020/04/14) was after the study period (March 30 to April 6, page 8). More details on the exact timing of protocol writing, ethics approval, study conduct, etc seem appropriate.

Reply: Please note that in the Supplementary Methods section, Study Design, we state that "This study was conducted between March 31st, 2020 and April 6th, 2020 in Gangelt...", and that in the DRKS00021306 entry, the first study participant is on March 31. Regarding the registry date, the reviewer raises an important point. We now specify in the Supplementary Methods section, Study Design (page 3, lines 18-22) that "The study was conducted in accordance with good clinical (GCP) and epidemiological practice (GEP) standards and the Declaration of Helsinki (2013), except that, due to time constraints in the situation of the pandemic, this epidemiological non-intervention study was registered April 14, 2020 shortly after the study period (March 31 to April 6, 2020)." Please note that to the best of our knowledge, other sero-epidemiological COVID-19 studies, such as some performed by the RKI, have not been registered at all, and registration not mandatory according to the WHO.

As to timing of protocol writing, ethics approval and study conduct: in the Supplementary Methods section, Study Design, we state "The study was approved by the Ethics Committee of the Medical Faculty of the University of Bonn (approval number 085/20)." The Ethics Committee was continuously informed about the research activities in Heinsberg from March 3 on and gave provisional approval for research activities in Heinsberg already on March 4. Revisions were approved March 24. The protocol and study materials of the sero-epidemiological study as presented in this manuscript were submitted to the Ethics Committee as an amendment to 085/20 on March 30, and approved March 31.

Reviewer 1: Page 6: "with regards to events4", should this refer to the WHO reference [5]?

Reply: Corrected.

Reviewer 1: page 8: I would suggest to drop the sentence "By definition, GEE models employ quasi-likelihood methods to obtain point estimates and CIs."

Reply: This sentence is now deleted as suggested.

Reviewer 1: page 8, "Statistical analysis was carried out by two independently working statisticians (MS, MB) (further details see suppl. methods)": Was there a statistical analysis plan (SAP)? If yes, then this could be mentioned here.

Reply: Due to time constraints in the pandemic, there was no SAP written before the analysis was conducted (please see above).

Reviewer 1: page 8, bottom: The reader might want to know the age of the 7 deaths (and of the 8th death mentioned in the discussion).

Reply: The mean age and the standard deviation of the 7 deaths is reported on page 9, line 25 and legend of Fig. 3C: (average age 80.8 years, sd \pm 3.5 years). For confidentiality reasons, the exact age of the 8th death (appr. 12,000 population in Gangelt) unfortunately cannot be provided in the manuscript due to the very high re-identification risk.

Reviewer 1: page 10 and following: I was wondering which age categorizations have been used for age-adjustments, there are different ones in Supp Mat (see my comment above).

Reply: Please see above.

Reviewer 1: Perhaps some comments might be appropriate in the Supp Mat that the odds ratio estimates from a logistic regression GEE are "population-averaged", so are ratios of population odds, not ratios of an individual's odds.

Reply: We added a respective sentence as suggested (page 4 lines 8-10).

Reviewer 1: I was pleased to see the Rogan and Gladen (1978) approach being used to adjust for misclassification. Is the uncertainty in sensitivity and specificity small enough to be ignored? Can you rule out that misclassification was differential? See e.g. K. J. Rothman, S. Greenland, and T. L. Lash. Validity in epidemiological studies. In K. J. Rothman, S. Greenland, and T. L. Lash, editors, Modern Epidemiology, pages 128–147. Lippincott Williams & Wilkins, Philadelphia, 3rd edition, 2008.

Reply: Thank you very much for mentioning this important aspect. Sensitivity and specificity values were obtained from the Anti-SARS-CoV-2-ELISA (IgG) manufacturer's validation data sheet (Euroimmun, version April 7th, 2020, see attachment 3). For example, specificity estimates were based on 1656 samples. The Anti-SARS-CoV-2-ELISA (IgG) and PCR tests are based on established laboratory procedures not involving issues such as recall bias. However, differential misclassification cannot be ruled out for some of the questionnaire data reported by study participants (e.g. regarding symptom variables and participation in carnival festivities, please see replies to Reviewer #2 below).

Reviewer #2

Reviewer 2: In this paper, authors perform a detailed investigation of an outbreak of SARS-CoV-2 in a small town in Germany, with a large super-spreading event that occurred during carnival festivities. A large serosurvey was performed to identify infected individuals along with results of PCR testing. Authors estimate that 15% of the population was infected, leading to an IFR of 0.36%. Participation in the carnival increased both the infection rate and the number of symptoms in the infected. Transmission in households was also quantified. This is a well-conducted and interesting study that provides important insights on the epidemiology of SARS-CoV-2. I have a couple of comments.

1) Dependency of the IFR estimate on the age of infected individuals: Since the severity of SARS-CoV-2 infection drastically increases with age, the estimate of the IFR of SARS-CoV-2 in a population is highly dependent on the proportion of elderly individuals among infected people. With a slight increase of that proportion, one could expect substantial increase in the population-level IFR. It appears that this outbreak was largely driven by a super-spreading event that occurred during carnival festivities. If younger members of the community were more likely to attend this even than elderly people, it is possible that infected people in this outbreak were younger than what they typically are in Germany. As a result, the IFR in this outbreak may be lower than what it would typically be in Germany. Authors need to discuss the dependency of the IFR estimate on the age distribution of infected individuals and how it may potentially lead to IFR in this community being lower than what would typically be expected in Germany. Authors should document the age distribution of those that attended the carnival and compare it to the age distribution of the population. Was age an important risk factor for carnival attendance? Do authors know the ages of individuals who died and could they provide age-specific estimates of IFR?

Reply: The reviewer is correct in assuming that the age of study participants that took part in carnival festivities is lower than the age of study participants not celebrating carnival, as demonstrated by our

newly performed analysis of carnival participation vs. age (41.42 ± 19.79 years vs. 56.93 ± 20.16 years on average, $p < 0.001$, see page 13 lines 8-12 of the revised manuscript). It is therefore possible that the age distribution of the infected persons in Gangelt is indeed different from the age distribution of infected in regions without a super-spreading event. Unfortunately, it is extremely difficult to obtain reliable data for the latter, as the percentages of infected individuals outside regions with super-spreading events in Germany are below 2 %, and testing strategies vary considerably. It is, however, possible to provide an age-specific estimate of the IFR in Gangelt: Since all individuals who died were older than 65 (please see page 9, line 25), this estimate is given by $7 /$ (estimated number of infected people in Gangelt > 65 years = $7 / 362 = 1.93\%$). This IFR estimate is clearly larger than the respective estimate for the whole population in Gangelt (0.36%). We now report this estimate on page 16 lines 14-16) and discuss the above aspects in more detail.

Reviewer 2: ad 2) Authors find that infected individuals who attended carnival festivities reported a larger number of symptoms. While this could indeed be due to a dose-response phenomenon, this might also potentially be explained by different reporting/recall rates. Indeed, if the carnival was quickly linked in the media to the outbreak, it is possible that people who attended the carnival were more quickly aware they were at risk and more prone to monitor and recall their symptoms. This possibility should be discussed.

Reply: Thank you for mentioning these important aspects. In fact, due to the use of questionnaire data, we cannot rule out a possible recall bias ("differential misclassification") in the symptom variables. We now discuss this limitation in the discussion section of the revised manuscript (page 16, lines 16-20).

Minor comments:

Reviewer 2: Page 3: "The current estimate of the CFR in Germany by WHO is between 2.2% and 3.4%". I think this may be a bit confusing for an audience of non-specialists. The CFR is often defined by the proportion of individuals with symptomatic infections who die; and no one believes this proportion is as high as 2-3%. It would be good if authors could provide the more narrow definition they use here and clarify the difference with the proportion of symptomatic infections who die upfront.

Reply: We agree that the CFR is a notoriously difficult parameter in a pandemic with numerous variables differing between countries. Furthermore, the calculated percentages rapidly change (currently the calculated CFR in Germany is approx. 4.1 % (Aug 18, 2020: 9,232 deaths / 224,014 deaths). Therefore, we changed the wording to: "Most estimates of the CFR (case fatality rate) are based on cases detected through surveillance and calculated using crude methods, giving rise to widely variable estimates of CFR by country as outlined by the WHO¹." The advantages of calculating the IFR is described in detail already in the Introduction section. For clarity, we have now added the calculation formula (page 4, lines 17-18): "...the infection fatality rate (IFR = number of deaths from disease / number of infected people) includes the whole spectrum of infected individuals, from asymptomatic to severe."

Reviewers' Comments:

Reviewer #1:

Remarks to the Author:

Thank you for a carefully conducted revision addressing all my comments.

Reviewer #2:

Remarks to the Author:

I'm happy with the last version of the manuscript.